# *wd1*: WEIGHTED POLICY OPTIMIZATION FOR REASONING IN DIFFUSION LANGUAGE MODELS

**Xiaohang Tang**[1,2*] **Rares Dolga**[2,5*] **Sangwoong Yoon**[3†] **Ilija Bogunovic**[4,2†]

[1]Department of Statistical Science, University College London, United Kingdom
[2]UCL AI Centre, University College London, United Kingdom
[3]Graduate School of AI, Ulsan National Institute of Science and Technology, South Korea
[4]Department of Mathematics and Computer Science, Universität Basel, Switzerland
[5]UIPath

## ABSTRACT

Improving the reasoning capabilities of diffusion-based large language models (dLLMs) through reinforcement learning (RL) remains an open problem. The intractability of dLLMs likelihood function necessitates approximating the current, old, and reference policy likelihoods at each policy optimization step. This reliance introduces additional computational overhead, and can lead to large variance and estimation error in RL objective – particularly in computing the policy ratio for importance sampling. To mitigate these issues, we introduce *wd1*, a novel ratio-free policy optimization approach that reformulates the RL objective as a weighted log-likelihood, requiring only a single approximation for the current parametrized policy likelihood. We formally show that our proposed method can be interpreted as energy-guided discrete diffusion training combined with negative sample unlearning, thereby confirming its theoretical soundness. In experiments on LLaDA-8B model, *wd1* outperforms diffusion-based GRPO (*d1*) while requiring lower computational cost, achieving up to a $+59\%$ improvement in accuracy. Furthermore, we extend *wd1* to denoising-stepwise weighted policy optimization (*wd1++*), achieving state-of-the-art math performance of $44.2\%$ on MATH500 and $84.5\%$ on GSM8K with only 20 RL training steps.

## 1 INTRODUCTION

Diffusion-based large language models (dLLMs) have recently gained attention as promising alternatives to autoregressive (AR) models for language modelling tasks (Nie et al., 2025b; Ou et al., 2025b; Yang et al., 2025). Unlike AR models, which generate tokens sequentially, dLLMs iteratively refine entire response sequences through a denoising process. A primary advantage of this approach is the significantly improved inference efficiency. Notably, recent closed models such as Mercury (Labs et al., 2025) and Gemini Diffusion achieve over an order of magnitude speed-up in generation compared to AR models, while maintaining comparable generation quality. Furthermore, open-weight dLLMs demonstrate competitive performance on standard language benchmarks, with smaller models (Lou et al., 2024; Ou et al., 2025b; Nie et al., 2024) achieving parity with equivalently sized AR baselines, and larger-scale models such as LLaDA-8B (Zhu et al., 2025a) and Dream-7B (Ye et al., 2025) extending this trend at scale. While dLLMs demonstrate strong performance in text generation, it remains an open and important question how best to fine-tune dLLMs using RL – a paradigm that has proven highly effective in alignment and improving reasoning capabilities of AR models (Ouyang et al., 2022; Shao et al., 2024).

A key challenge in applying reinforcement learning (RL) to dLLMs is the intractability of their likelihood functions (Zhao et al., 2025; Yang et al., 2025), which necessitates approximation for policy optimization. Applying approximated log-likelihood for diffusion-based GRPO (Shao et al., 2024; Zhao et al., 2025) can exponentially amplify the approximation error and lead to large variance

---

*Equal contribution. Code: https://github.com/xiaohangt/wd1
†Corresponding authors (ilija.bogunovic@unibas.ch, swyoon@unist.ac.kr).

when computing the policy ratio for importance sampling. Moreover, GRPO requires the estimated likelihoods of the current, old, and reference policies at every training step, leading to significant computational overhead. These issues can be further exacerbated as the completion length and the number of diffusion steps increase.

To address these challenges, we propose *wd1*, a policy optimization approach with **w**eighted log-likelihood objective for **d**LLMs. Crucially, this objective dispenses with explicit policy ratios and relies on a single likelihood approximation, thereby avoiding the potentially large bias and variance in policy ratio, and reducing the computational overhead. Our principal contributions are:

- We propose a novel reinforcement learning method for dLLMs, termed *wd1*, which formulates the objective as a weighted log-likelihood of outcome sequence, derived from reverse-KL regularized policy optimization. The weight, defined as $(-w^+ + w^-)$ and dependent on the advantage $A$, balances two terms: $w^+ \propto \exp(A)$ increases the probability of higher-advantage completions, while $w^- \propto \exp(-A)$ decreases the probability of lower-advantage ones. Together, this mechanism amplifies beneficial outcomes meanwhile actively reducing detrimental ones.

- We prove that our proposed RL method for dLLMs can be interpreted as jointly training an energy-guided discrete diffusion model—guided by the advantage function—and unlearning low-advantage data, thereby steering generation toward higher-advantage completions.

- We conduct experiment with LLaDA-8B-Instruct model (Nie et al., 2025a). Compared to the baseline method *d1* (Zhao et al., 2025), our method *wd1* achieves **76.4% on Sudoku (Arel, 2025) (+58.8% over *d1*)** and **51.2% on Countdown (Pan et al., 2025) (+16% over *d1*)**, without requiring supervised fine-tuning (SFT), and with significantly less computational burden in RL training.

- We further extend our method to leverage intermediate completions generated in the decoding process, which we call *wd1++*. The extended method surpasses several concurrent RL for dLLMs methods, achieving state-of-the-art performance **44.2% on MATH500** and **84.5% on GSM8K** with only 20 training steps, and $10\times$ fewer rollouts compared to the baseline methods.

## 2 PRELIMINARIES

We denote the generation policy of diffusion-based Large Language Models (dLLMs) by $\pi_\theta$. Denote prompt $q \in \mathcal{D}$, and completions $o \in \mathcal{O}$. Notably, the reward function denoted by $R(q, o)$ in this work is not limited to verifiers. We use superscript $k$ to indicate the $k$-th token of completion: $o^k$ or $x_0^k$.

### 2.1 DIFFUSION LARGE LANGUAGE MODELS

The prevailing class of discrete diffusion models for language modeling is masked diffusion models (MDMs), which gradually corrupt text sequences by replacing tokens with a special mask token (Lou et al., 2024; Shi et al., 2024; Sahoo et al., 2024; Ou et al., 2025b). Let $t \in [0, 1]$ denote the diffusion timestep, and $x_t$ as the masked sequence at step $t$. The fully denoised sequence (i.e., the completion $o$) is represented by $x_0$, and the forward process $(p_{t|0}(x_t \mid x_0))$ is formulated as a continuous-time Markov chain. The transition kernel $\mathbf{Q}_t$ is absorbing (Campbell et al., 2022; Austin et al., 2023), meaning that at time $t$, $\mathbf{Q}_t = \sigma(t)\mathbf{Q}^{\text{absorb}}$, where $\sigma$ is a decreasing scalar noise schedule and $\mathbf{Q}^{\text{absorb}}$ is a constant matrix (See Definition 2).

This work aims to apply reinforcement learning to fine-tune masked discrete diffusion models such as LLaDA (Ou et al., 2025b; Zhu et al., 2025a), which models the clean data distribution conditional on masked sequence as $\pi_\theta(x_0^k \mid x_t)$. A standard training objective for MDMs is the negative evidence lower bound (ELBO) in Denoising Cross Entropy (DCE) (Ou et al., 2025b) and MD4 (Shi et al., 2024): let $K$ denote the length of the sequence, $x_0^k$ denote the $k$-th token of $x_0$, $\forall x_0 \sim p_{\text{data}}$,

$$\mathcal{L}(x_0) = -\mathbb{E}_{t\sim\mathcal{U}[0,1],\ x_t\sim p_{t|0}(x_t|x_0)} \left[ \frac{1}{t} \sum_{k=1}^{K} \mathbf{1}(x_t^k = [\texttt{mask}]) \log \pi_\theta(x_0^k \mid x_t) \right], \qquad (1)$$

Specifically, the intermediate timestep $t$ is sampled uniformly, and the masked sequence $x_t$ is generated according to the predefined forward process $p_{t|0}(x_t \mid x_0)$. The resulting ELBO objective $\mathcal{L}$ is then commonly used as a tractable surrogate for the log-likelihood $\log \pi_\theta(x_0)$, enabling both supervised fine-tuning and reinforcement learning for MDMs (Nie et al., 2025a; Zhu et al., 2025a; Yang et al., 2025; Ou et al., 2025a).

## 2.2 EXISTING POLICY OPTIMIZATION METHODS

The base method of current prevailing RL fine-tuning algorithms is Trust Region Policy Optimization (TRPO) (Schulman et al., 2015), in which *forward* KL divergence is applied to restrict the update:

$$\max_{\theta} \ \mathbb{E}_{q\sim\mathcal{D}, \ o\sim\pi_\theta(\cdot|q)}\left[A^{\pi_{\text{old}}}(q, o) - \lambda D_{\text{KL}}\left(\ \pi_{\text{old}}(\cdot|q) \ \| \ \pi_\theta(\cdot|q)\ \right)\right], \tag{2}$$

where $A^{\pi_{\text{old}}}$ is the advantage function, $q$ and $o$ are denoted as the prompt and (clean) response, respectively. Proposition 1 (Appendix A) demonstrates the monotonic policy improvement of TRPO.

PPO then extends the soft constraint (KL penalty) to clipping policy ratio $\pi_\theta(\cdot|q)/\pi_{\text{old}}(\cdot|q)$ and employing pessimism for policy update, further employed in fine-tuning (Ouyang et al., 2022) with additional reverse-KL regularization w.r.t. the reference policy $\pi_{\text{ref}}$. Group Relative Policy Optimization (GRPO) (Shao et al., 2024) simplifies PPO by sampling a group of completions $\{o_i\}_{i=1}^G$ and approximating their advantage with their normalized rewards. This advantage is corrected by subtracting the mean reward across the group (Liu et al., 2025): $\hat{A}_i = R(q, o_i) - \texttt{mean}\big(R(q, o_{1:G})\big)$, which we refer to as the *group-relative advantage*.

## 2.3 POLICY OPTIMIZATION FOR DLLMS

Adapting GRPO to diffusion-based large language models (dLLMs) presents notable challenges, since dLLMs generate outputs via a non-autoregressive, iterative denoising process, making the computation of $\log \pi_\theta(o|q)$ intractable and necessitating approximation for policy optimization.

Existing works by Nie et al. (2025a); Yang et al. (2025) employ ELBO for per-token log-likelihood approximation following DCE: $\phi^\pi(x_0^k) = \mathbb{E}_{t\in\mathcal{U}[0,1]}[w \cdot \mathbf{1}[x_t^k = \texttt{mask}] \log \pi(x_0^k|x_t, q)]$, where $w = 1/t$ in DCE and $w = 1$ in UniGRPO (Yang et al., 2025). However, an accurate estimation requires a large sample size of $t$, resulting in inefficiency for online RL. A biased but efficient method is introduced in *d1* (Zhao et al., 2025), requiring only sample at $t = 1$: $\phi^\pi(x_0^k) = \log \pi(x_0^k|x_1, q')$, where prompt $q'$ is randomly masked, $x_1$ is fully masked response.

In diffusion-based GRPO (Zhao et al., 2025; Yang et al., 2025), policy ratio is then computed using the approximated log-likelihoods: $r_i^k(\theta) = \pi_\theta(o_i^k)/\pi_{\text{old}}(o_i^k) \approx \exp\big(\phi^{\pi_\theta}(o_i^k) - \phi^{\pi_{\text{old}}}(o_i^k)\big)$ for importance sampling in estimating the objective of GRPO:

$$\mathbb{E}_{\substack{q\sim\mathcal{D}, \\ o_{1:G}\sim\pi_{\text{old}}(\cdot|q)}}\left[\frac{1}{GK}\sum_{i=1}^{G}\sum_{k=1}^{K}\min\big(r_i^k(\theta)\hat{A}_i, \text{clip}(r_i^k(\theta), 1\pm\epsilon)\hat{A}_i\big) - \beta D_{\text{KL}}\big(\pi_\theta(\cdot) \| \pi_{\text{ref}}(\cdot)\big)\right]. \tag{3}$$

However, existing approaches are hampered by their reliance on extensive likelihood approximation to compute the policy ratio. In current diffusion-based GRPO methods, the ratio is computed as $r_i^k \approx \exp\big(\phi^{\pi_\theta}(o_i^k) - \phi^{\pi_{\text{old}}}(o_i^k)\big)$ so approximation errors in likelihood can be *exponentially* amplified. As formally shown in Appendix A.1, the resulting error in the estimated RL objective becomes more severe when less accurate log-likelihood approximations are used, such as in *d1*, or ELBO used in DCE and Uni-GRPO when the Monte Carlo sample size $t$ is small.

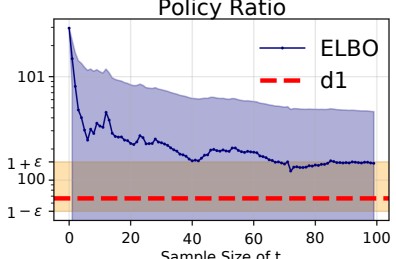

Figure 1: Example policy ratio value $r_i^k$ computed using ELBO and approximated likelihood in *d1* on GSM8K after a policy update. Ratio's unclipped interval is $[1 - \epsilon, 1 + \epsilon]$, where $\epsilon = 0.5$. ELBO-based likelihood approximation yields high-variance ratio estimates; *d1* induces a biased ratio that can deviate substantially from ELBO. Both methods suffer from efficiently and accurately compute ratios.

Although increasing $t$ in the ELBO estimator can reduce approximation error, the induced ratio estimates can still exhibit high variance, as illustrated in Figure 1. Although alternative approximator such as that in *d1* can improve efficiency, but yields a biased ratio that can differ substantially from the ELBO-based ratio, thereby introducing a systematic bias into the RL training objective. Finally, GRPO requires applying the approximation function $\phi$ separately to three policies—$\pi_\theta$, $\pi_{\text{old}}$, and $\pi_{\text{ref}}$—which further increases computational overhead.

# 3 *wd1*: WEIGHTED POLICY OPTIMIZATION FOR DLLMS

In this section, we introduce *wd1*, a novel RL algorithm that eliminates the need for approximating the likelihood (policy) ratios for importance sampling, aiming to reduce the computational burden, and the variance and approximation error in computing the RL objective. We further extend our method to *wd1++* by applying denoising-stepwise policy optimization.

## 3.1 REINFORCEMENT LEARNING AS WEIGHTED LOG-LIKELIHOOD MAXIMIZATION

Prevailing RL methods are based on constrained policy optimization (Belousov & Peters, 2017), penalizing the deviation of current policy $\pi_\theta(\cdot|q)$ from the old policy $\pi_{\text{old}}(\cdot|q)$. TRPO objective (Equation (2)) applies a forward-KL penalty. We instead adopt reverse-KL penalty augmented with the reference policy regularization $D_{\text{KL}}(\pi_\theta(\cdot|q) \,\|\, \pi_{\text{ref}}(\cdot|q))$:

$$\max_\theta \ \mathbb{E}_{q \in \mathcal{D}, o \sim \pi_\theta(\cdot|q)}\Big[A^{\pi_{\text{old}}}(q,o) - \lambda D_{\text{KL}}\Big(\ \pi_\theta(\cdot|q) \,\|\, \pi_{\text{old}}(\cdot|q)\ \Big) - \beta D_{\text{KL}}\Big(\pi_\theta(\cdot|q) \,\|\, \pi_{\text{ref}}(\cdot|q)\Big)\Big]. \tag{4}$$

Note that the monotonic improvement guarantee still holds when using reverse-KL penalty, as we show in Theorem 2. From the method of Lagrange multipliers, the solution to Equation (4) has the following form (Peng et al., 2019; Rafailov et al., 2023):

$$\pi^*(\cdot|q) \propto \pi_{\text{old}}(\cdot|q)^{\lambda/(\lambda+\beta)} \cdot \pi_{\text{ref}}(\cdot|q)^{\beta/(\lambda+\beta)} \cdot \exp\left(\frac{A^{\pi_{\text{old}}}(q,\cdot)}{\lambda+\beta}\right). \tag{5}$$

As the analytic form of the optimal policy $\pi^*$ is known, we can train our policy by directly minimizing $D_{\text{KL}}(\pi^*(\cdot|q) \,\|\, \pi_\theta(\cdot|q))$. This minimization can be expressed as the following weighted log-likelihood (WLL) loss, where the weights $\propto \exp\left(\psi A^{\pi_{\text{old}}}\right)$, $\psi = \frac{1}{\lambda+\eta}$ and the samples are obtained from the geometric mixture policy $\pi_{\text{old}}^{\text{ref}}(\cdot|q) \propto \pi_{\text{old}}(\cdot|q)^{\lambda/(\lambda+\beta)} \cdot \pi_{\text{ref}}(\cdot|q)^{\beta/(\lambda+\beta)}$ (See Proposition 2): $\forall q \sim \mathcal{D}$,

$$\mathcal{L}_{\text{WLL}}(\theta) = \mathbb{E}_{o \sim \pi_{\text{old}}^{\text{ref}}(\cdot|q)}\Big[ -\exp\left(\psi A^{\pi_{\text{old}}}(q,o)\right) \cdot \log \pi_\theta(o|q)\Big] \tag{6}$$

$$\approx \mathbb{E}_{\{o_i\}_{i=1}^G \sim \pi_{\text{old}}^{\text{ref}}(\cdot|q)}\left[\frac{1}{G}\sum_{i=1}^G -\frac{\exp\left(\psi \hat{A}_i\right)}{\sum_{j=1}^G \exp\left(\psi \hat{A}_j\right)} \log \pi_\theta(o_i|q)\right]. \tag{7}$$

As shown in Equation (7), we approximate the advantage function using the group-relative advantage $\hat{A}$ and normalize the weights, thereby limiting their magnitude and reducing variance in loss computation. Notably, dividing by the normalization constant does not affect the solution, since it is independent of the completions. The resulting objective does not involve ratio $\pi_\theta(\cdot|q)/\pi_{\text{old}}(\cdot|q)$ for importance sampling or $\pi_\theta(\cdot|q)/\pi_{\text{ref}}(\cdot|q)$ for regularization, successfully avoiding the potential amplification of log-likelihood approximation error and large variance in diffusion GRPO.

Although the objective $\mathcal{L}_{\text{WLL}}(\theta)$ in Equation (7) avoids the likelihood ratio estimation, it has two limitations. First, the algorithm is not fully utilizing all the completions. Due to the exponential form of the weighting, completions with relatively low advantage – referred to as *negative* samples – may receive vanishingly small weights, and do not contribute to learning. Second, due to the likelihood-maximization property of WLL, the likelihoods of all sampled completions are increased, even for negative samples. This issue is exacerbated in scenarios where all completions attain identical but low rewards (e.g. 0), thus all weights become equal and the likelihoods of these suboptimal samples are nonetheless reinforced.

## 3.2 *wd1*: FULLY UTILIZING COMPLETIONS

We propose *wd1*, an improved weighted log-likelihood objective that explicitly reinforces positive samples and penalizes negative samples:

$$\mathcal{L}_{wd1}(\theta) = \mathbb{E}_{q \sim \mathcal{D}, \{o_i\}_{i=1}^G \sim \pi_{\text{old}}^{\text{ref}}(\cdot|q)}\left[\frac{1}{G}\sum_{i=1}^G \big(-w^+(q,o_i) + w^-(q,o_i)\big) \cdot \log \pi_\theta(o_i|q)\right], \tag{8}$$

where the weights are based on group-relative (GRPO) advantage and are further normalized to avoid overly imbalanced weight $\hat{A}_i = R(q, o_i) - \texttt{mean}(R(q, o_{1:G}))$:

$$w^+(q, o_i) = \frac{\exp\left(\psi\hat{A}_i\right)}{\sum_{j=1}^{G}\exp\left(\psi\hat{A}_j\right)}, \quad w^-(q, o_i) = \frac{\exp\left(-\psi\hat{A}_i\right)}{\sum_{j=1}^{G}\exp\left(-\psi\hat{A}_j\right)}. \tag{9}$$

*wd1* objective balances positive and negative samples through a complementary penalty term, $w^-(q, o_i)\log\pi_\theta(o_i|q)$, which minimizes the likelihood of low-advantage completions. This penalty induces negative gradients, thereby accelerating divergence from undesirable completions. Moreover, in the extreme case where all completions exhibit identical advantages, the optimization naturally halts since $w^+ = w^-$, thereby addressing the concern on increasing likelihood of negative samples proposed in Sec 3.1. We demonstrate the effectiveness of this combination via ablations in C.2.

Our method *wd1*, a simple ratio-free policy optimization based on **w**eighted log-likelihood objective for **d**LLMs, is formally presented in Algorithm 1. We first obtain $G$ completions $\{o\}_{i=1}^{G}$ sampled from geometric mixture $\pi_{\text{old}}^{\text{ref}}(\cdot|q) \propto \pi_{\text{old}}(\cdot|q)^{\lambda/(\lambda+\beta)} \cdot \pi_{\text{ref}}(\cdot|q)^{\beta/(\lambda+\beta)}$ (line 5). Since the base model LLaDA parametrizes the clean token prediction $\pi_{\text{old}}^{\text{ref}}(x_0^k|x_t, q)$ for denoising, we approximate $\log\pi_{\text{old}}^{\text{ref}}(x_0^k|x_t, q) \approx \lambda\log\pi_{\text{old}}(x_0^k|x_t, q) + \beta\log\pi_{\text{ref}}(x_0^k|x_t, q)$ as the logits of the denoising distribution at each step $t$. We then use the samples to compute weights in Equation (9) (line 6). In weights computing, we leverage completions from all groups to estimate the normalization constant, in order to restrict the the gradient norm and stabilize the training. Finally in line 8, we approximate the log-likelihood of completions, and compute objectives for policy update. Likelihood approximation in *d1* (Zhao et al., 2025) is directly applicable to *wd1*: $\log\pi_\theta(x_0|q) \approx \sum_k \log\pi_\theta(x_0^k|x_1, q')$, where $q'$ is randomly masked from prompt $q$ at every gradient step.

### 3.3 *wd1++*: STEPWISE WEIGHTED POLICY OPTIMIZATION

The decoding process in dLLMs relies on confidence-based remasking (Wang et al., 2025b). At each denoising step $l \in \{1, \cdots, L\}$ in decoding, clean data is predicted conditional on the masked sequence $x_l$ and then tokens with low-confidence are re-masked for further denoising, which construct a refinement process. Since current diffusion RL methods only use the final predicted clean completion for training, there are bunch of clean completions in the intermediate denoising steps remaining *unused*.

To leverage intermediate clean completions, we extend our weighted log-likelihood objective to a step-wise formulation based on DCE, which we term *wd1++*. In *wd1* (as well as in GRPO), a group of completions $\{o_i\}_{i=1}^{G}$ is sampled for policy optimization. In *wd1++*, we expand this group to $\{O_i\}_{i=1}^{G}$, where $O_i = \{x_{0|l} \mid x_{0|l} \sim \pi_{\text{old}}^{\text{ref}}(\cdot \mid x_t, q), \ x_{0|L} = o_i, \ l = 1, \ldots, L\}$, which contains all generated completions during the decoding process, including intermediate ones. The expanded group of completions is then used to estimate both the advantage function and the corresponding weights. The resulting loss objective is defined as:

$$\mathcal{L}_{wd1++}(\theta) = \mathbb{E}_{\substack{q \sim \mathcal{D}, \\ \{O_i\}_{i=1}^{G} \sim \pi_{\text{old}}^{\text{ref}}(\cdot|q) \\ l \in \text{Unif}\{1, \cdots, L\}}} \left[ \frac{L}{Gl} \sum_{i=1}^{G} \sum_{x_{0|l} \in O_i} \left( -w^+(q, x_{0|l}) + w^-(q, x_{0|l}) \right) \cdot \log\pi_\theta(x_{0|l}|x_l, q) \right]. \tag{10}$$

## 4 THEORETICAL INSIGHTS: ENERGY-GUIDED DIFFUSION SAMPLING

In this section, we present a novel theoretical interpretation of policy optimization for *dLLMs*. We prove that the advantage-weighted log-likelihood objective (*wd1*) for dLLMs can be viewed as energy-guided discrete diffusion training combined with negative sample unlearning.

Sampling from the solution policy of the reverse-KL policy optimization, as described in Equation (5), can be interpreted as energy-guided sampling, where the energy function $\mathcal{E}(q, \cdot) = -A^{\pi_{\text{old}}}(q, \cdot)$. Equation (5) defines the marginal distribution of the clean data ($x_0 = o$) which we denote as $p_0^*(x_0)$[1]. To obtain the guidance at intermediate time steps $t > 0$, we define the forward diffusion process for the target diffusion policy $\pi^*$ as following.

**Definition 1.** *The forward diffusion process of the target policy ($\pi^*$) satisfies $p_{t|0}^*(x_t|x_0) = p_{t|0}(x_t|x_0)$, where $p_{t|0}$ is the forward process of old diffusion policy $\pi_{\text{old}}$.*

---

[1]To adapt to the setting of diffusion, we use $x_t$ to denote the (masked) completions, and omit the prompt $q$.

---

**Algorithm 1** *wd1*: **W**eighted Policy Optimization for **dLLMs**

---

**Require:** Reference model $\pi_{\text{ref}}$, prompt distribution $\mathcal{D}$, group size $G$, reward function $R$, dLLM $\pi_\theta$, regularization hyperparameters $\lambda$ and $\beta$

1: Initialize $\pi_\theta \leftarrow \pi_{\text{ref}}$
2: **while** not converged **do**
3:     $\pi_{\text{old}} \leftarrow \pi_\theta$
4:     Sample prompt $q \sim \mathcal{D}$ and $G$ completions $o_i \sim \pi_{\text{old}}(\cdot \mid q), \forall i \in [G]$
5:     Compute advantage $\hat{A}_i = R(q, o_i) - \texttt{mean}(R(q, o_{1:G})), \forall i \in [G]$
6:     Compute weights $w^+$ and $w^-$ in Equation (9), $\forall i \in [G]$
7:     **for** gradient update iterations $n = 1, 2, \ldots, \mu$ **do**
8:         Compute approximated log-likelihood $\log \pi_\theta(o_i|q)$
9:         Compute objective $\mathcal{L}_{wd1}(\theta)$ in Equation (8) or Equation (10) and update $\theta$
10:     **end for**
11: **end while**
12: **return** $\pi_\theta$

---

Since the reference diffusion policy is the initial policy, three policies have *identical forward diffusion process*, being $p^*_{t|0}(x_t|x_0) = p_{t|0}(x_t|x_0) = p^{\text{ref}}_{t|0}(x_t|x_0)$, and thus, $p^*_{t|0}(x_t|x_0) = p'_{t|0}(x_t|x_0)$, where $p'$ is the geometric mixture diffusion $p'_{t|0}(x_t|x_0) \propto p_{t|0}(x_t|x_0)^\lambda p^{\text{ref}}_{t|0}(x_t|x_0)^\beta$. We can then obtain the energy guidance at all time step $t$.

**Lemma 1** (Intermediate Energy Guidance on Discrete Diffusion). *The marginal probability distribution of the masked responses $(x_t)$ in the diffusion process satisfies $p^*_t(x_t) = p'_t(x_t) \cdot \exp\big(A_t(x_t)\big)/Z_t$, which induces an energy-guided discrete diffusion:*

$$p^*_{0|t}(x_0|x_t) \propto p'_{0|t}(x_0|x_t) \cdot \exp(A(x_0) - A_t(x_t)), \tag{11}$$

*where $-A_t(x_t) = -\log \mathbb{E}_{x_0 \sim p'_{0|t}(\cdot|x_t)}[\exp\big(A(x_0)\big)]$ is intermediate energy function for $t > 0$, and $A(\cdot)$ is advantage function (Proof in Appendix A.3).*

The guidance provided in Lemma 1 demonstrates that it directs the sampling process toward generating completions that exhibit higher advantage values. However, conducting training-free guided sampling following Equation (11) requires estimating the posterior mean of the exponential of advantage (Lu et al., 2023). Rather than relying on such estimation, we instead aim to find the training objective to directly approximate the target guided diffusion model.

Since existing masked dLLMs parametrize the concrete score (Meng et al., 2022), to apply the energy guidance, we aim to directly approximate target guided concrete score. Denote $x_t = (x^1_t, \cdots, x^d_t)$ and $\hat{x}_t$ is identical to $x_t$ except the $i$-th token is unmasked (i.e. $x^i_t = [M]$ and $\hat{x}^i_t \neq [M]$). Concrete score is defined as the marginal probability ratio between $\hat{x}_t$ and $x_t$:

$$s(x_t, t) \stackrel{\text{def}}{=} \frac{p(x^1_t, \cdots, \hat{x}^i_t, \cdots, x^d_t)}{p(x^1_t, \cdots, x^i_t, \cdots, x^d_t)}. \tag{12}$$

We prove that the training objective to approximate the guided concrete score can be simplified as a weighted Denoising Concrete Score Matching (D-CSM) (Meng et al., 2022):

**Theorem 1.** *The model $s_\theta$ approximates the concrete score of the energy-guided discrete diffusion $p^*$ when the following loss objective is minimized. This objective is in a form of **advantage-weighted** Denoising Concrete Score Matching, which we call AW-D-CSM:*

$$\mathcal{L}_{\textit{AW-D-CSM}} = \mathbb{E}_{x_0 \sim p'_0(\cdot)}\bigg[\underbrace{\exp\big(A(x_0)\big)}_{\textit{Advantage Weight}} \cdot \underbrace{\mathbb{E}_{t \sim [0,T], p'_{t|0}(x_t|x_0)}[\|s_\theta(x_t, t) - \frac{p'_0(\hat{x}_t|x_0)}{p'_0(x_t|x_0)}\|^2_2]}_{\mathcal{L}_{\textit{D-CSM}}(x_0)}\bigg]. \tag{13}$$

We provide the proof in Appendix A.3. Additionally, D-CSM is an approximation of CSM (Meng et al., 2022), which is equivalent to Denoising score entropy (DSE) (Lou et al., 2024). For all $x_0$, it is satisfied up to multiplying a constant that $\mathcal{L}_{\text{D-CSM}}(x_0) \Leftrightarrow \mathcal{L}_{\text{CSM}}(x_0) \Leftrightarrow \mathcal{L}_{\text{DSE}}(x_0) \Leftrightarrow \mathcal{L}_{\text{DCE}}(x_0)$

(Ou et al., 2025b). Therefore, AW-D-CSM can then be applied for both SEDD (Lou et al., 2024) and RADD (Ou et al., 2025b) model such as LLaDA. Denote $p_\theta$ as the concrete score reparametrized model, AW-D-CSM can be converted to a weighted denoising cross-entropy loss (AW-DCE):

$$\mathcal{L}_{\text{AW-DCE}} = \mathbb{E}_{x_0 \sim p'_0(\cdot)} \left[ \exp\left(A(x_0)\right) \cdot \mathbb{E}_{t \sim [0,T], p'_{t|0}(x_t|x_0)} \left[ \sum_{x_t^i = [\text{mask}]} -\frac{1}{t} \log p_\theta(x_0^i | x_t^{\text{UM}}) \right] \right]. \quad (14)$$

DSE and DCE objectives both can be used for likelihood approximation in fine-tuning (Ou et al., 2025b; Nie et al., 2025a; Yang et al., 2025) since they can serve as negative ELBO (Lou et al., 2024; Shi et al., 2024). Thus, the advantage-weighted objective AW-DCE (or AW-DSE) used to learn energy-guided score is in a weighted log-likelihood form as in *wd1* with only $w^+$ (i.e. WLL loss in Equation (6)), which contributes to our main theoretical findings:

**Remark 1.** *In the context of applying RL to masked discrete diffusion, the advantage-weighted log-likelihood (WLL) objective (Equation (6)) induced by reverse-KL policy optimization, is equivalent to the objective of training energy-guided diffusion models, where the energy function is the negative advantage. Formally, $\mathcal{L}_{WLL} \Leftrightarrow \mathcal{L}_{AW\text{-}DCE}$ when DCE is used for likelihood approximation.*

**Remark 2.** *Additionally, based on DCE likelihood, the additional penalty term on negative samples used to extend WLL to wd1 loss can be viewed as applying data unlearning by minimizing the ELBO (Alberti et al., 2025), where the data $\{x_0^-\}$ (negative samples) has probability distribution $p_{data}(x_0^-) \propto p'_0(x_0^-) \exp(-A(q, x_0^-))$, which corresponds to a Boltzmann distribution that places higher probability mass on regions with lower advantage values (more details in Appendix D.1).*

## 5 EXPERIMENTS

In this section, we empirically validate the following key advantages of our approach:

  i)  Improved reasoning capabilities than existing methods on popular reasoning benchmarks;
 ii)  reduced computational burden, as reflected by decreased runtime, lower FLOPs and numbers of function evaluations (NFEs) per training step, number of training steps and rollouts; and
iii)  marked performance gains attributable to the incorporation of samples with low-advantage.

To evaluate our approach, we next detail the experimental setup and implementation.

**Experimental Setup.** We perform reinforcement learning (RL) fine-tuning on the LLaDA-8B-Instruct model (Nie et al., 2025a) with Low-Rank Adaptation (LoRA) on: GSM8k (Cobbe et al., 2021), MATH (Lightman et al., 2023), Sudoku (Arel, 2025), and Countdown (Pan et al., 2025). As for decoding, we follow the default strategy Mounier & Idehpour (2025); Arriola et al. (2025); Wang et al. (2025b). Our main baseline is *d1* (Zhao et al., 2025), the *first* RL method developed for masked diffusion LLMs (dLLMs). We reproduce the baseline methods *Diffu*-GRPO, which applies diffusion-based GRPO training directly to the LLaDA base model, and *d1*, which performs SFT before applying *Diffu*-GRPO. We use s1K (Muennighoff et al., 2025) data for SFT in *d1*. We also compare with SDPO (Han et al., 2025), TCR (Wang et al., 2025d), and MDPO (He et al., 2025) on benchmarks GSM8K and MATH500. MDPO is reproduced based on the official implementation and the training dataset (He et al., 2025).

**Implementation.** As for *wd1*, we conduct training on the same dataset as in *d1* (Zhao et al., 2025): training splits on GSM8k and MATH, and the dataset splits provided by Zhao et al. (2025) on Sudoku and Countdown. In our implementation of *wd1*, we apply the same likelihood approximation method as *d1*. The hyperparameters used in our method and our reproduction of *d1* are listed in Table 6 and Table 5. As for *wd1++*, we train on dataset provided by (He et al., 2025), which is sampled from OpenR1 dataset (Face, 2025). Since previous works (Yu et al., 2025) have demonstrated that the reference policy is empirically unnecessary, we set $\beta = 0$ and $\lambda = 1$ to eliminate $\pi_{\text{ref}}$ in practice. We report results using *zero-shot* and pass@1 evaluation on sequence lengths of 256 and 512 tokens.

### 5.1 MAIN RESULTS

**Superior Reasoning Ability.** In Table 1, we observe that *wd1*, even without supervised fine-tuning or using any supervised data, consistently outperforms our reproduced implementation of *d1*. Notably,

---

[2]In the technical report version of this work, our method achieved scores of 25.2 and 24.2 on Sudoku after 5K training steps. In this paper, we extend the training to 12.5K steps, and *wd1* results in improved performance.

Table 1: Test Accuracy (%) of **wd1** and *d1*. We reproduce *d1* and vary completion length. Our approach without SFT, demonstrates particularly higher accuracy on Sudoku[2]and Countdown.

| Model / Gen Len | Sudoku 256 | Sudoku 512 | Countdown 256 | Countdown 512 | GSM8K 256 | GSM8K 512 | MATH500 256 | MATH500 512 |
|---|---|---|---|---|---|---|---|---|
| LLaDA-8B-Instruct | 6.7 | 5.5 | 19.5 | 16.0 | 76.7 | 78.2 | 32.4 | 36.2 |
| + *diffu*-GRPO | 16.1 | 11.7 | 27.0 | 34.0 | 80.7 | 79.1 | **34.4** | **39.0** |
| + SFT + *diffu*-GRPO (*d1*) | 17.6 | 16.2 | 25.8 | 35.2 | 78.2 | 82.0 | **34.4** | 38.0 |
| + **wd1** | **76.4** | **62.8** | **51.2** | **46.1** | **80.8** | **82.3** | **34.4** | **39.0** |

Table 2: Comparison of Training Cost on 4×A100. We show SFT cost, average training time, FLOPs evaluated by DeepSpeed Flops Profiler, and theoretical NFEs per training step which includes $\mu = 8$ gradient steps. *wd1* removes SFT and has less cost per-step in RL than *d1*.

| Method | SFT Time Cost | RL Training Time Cost | RL Training FLOPs | RL Training NFEs for Likelihood |
|---|---|---|---|---|
| *d1* | 2.01 hrs | 103.5 sec/step | $9.922 \times 10^{15}$/step | $(\mu + 2)$/step |
| **wd1** | 0 hrs | 81.16 sec/step | $8.887 \times 10^{15}$/step | $\mu$/step |

*wd1* surpasses *d1* by 43% in test accuracy on the Sudoku task, and achieves up to a 25% improvement on Countdown with maximum length 256. Relative to the base LLaDA model, the performance gain reaches as high as 54% on Sudoku and 42% on Countdown. On math problem-solving benchmarks GSM8K and MATH500, *wd1* attains slightly higher accuracy. Nevertheless, the extended method *wd1++* obtains significantly better accuracy. In Table 3 (left), we further compare with concurrent baselines released in recent months. *wd1++* outperforms the baselines including strong one MDPO.

**Reduced Training Cost.** Table 2 demonstrates that the training cost required by *wd1* is substantially lower than that of *d1*. Unlike *d1*, *wd1* does not require a SFT stage, which alone accounts for approximately two hours of training in *d1*. *wd1* achieves additional speedup during the RL phase, where runtime is measured by averaging over $\mu = 8$ inner gradient steps per global step. Notably, the time efficiency gap is expected to widen further under settings with larger maximum sequence lengths and more diffusion steps. This efficiency gain is further supported by a reduced FLOPs and number of function evaluations (NFEs) per step, as *wd1* bypasses the need to approximate the likelihood of the old policy. We exclude NFEs associated with sampling, since both methods share identical sampling costs as *wd1* removes the reference policy regularization.

In Table 3 (right), we report the training cost required to obtain the best post-trained models on GSM8K and MATH500, measured in terms of the number of training steps and rollouts. *wd1++* requires $10\times$ fewer rollouts to achieve superior performance, clearly demonstrating the efficiency of our method. This rapid convergence arises primarily from the *exponential* advantage weights applied to the log-likelihood in *wd1*. In contrast, standard RL methods such as GRPO and PPO weight the log-likelihood (or policy ratio) terms directly by the advantage function.

## 5.2 ABLATION STUDY

We present an ablation study in Figure 4. Notably, we observe that supervised fine-tuning (SFT) yields only marginal improvements within our approach, with a slight gain in the Sudoku task. This contrasts with *d1*, where SFT plays a significant role in improving performance. These findings indicate that *wd1* can eliminate the need for an SFT phase, thereby simplifying the training pipeline and substantially reducing computational cost. Additionally, we evaluate the impact of removing the negative-weighted term by setting $w^- = 0$, thus relying solely on the positive advantage weights $w^+$. We provide further ablation on the combined method between $w^+$ and $w^-$ in Table 9. The results highlight the importance of explicitly penalizing the likelihood of low-advantage completions, thereby reinforcing the role of negative samples, and emphasize the critical balance between the two weights.

---

[3]To facilitate efficient ablation studies, we restrict our comparisons to checkpoints saved prior to 5000 steps.

Table 3: **Left:** Extended method *wd1++* compared to concurrent RL methods to fine-tune LLaDA-8B-Instruct. Methods denoted by "(full)" perform full fine-tuning. **Right:** Training cost to obtain the best model on GSM8K and MATH500. We count the total number of steps of policy iteration (model weights update), and the number of rollouts used for training (see Table 8 for details on counting).

| Model | GSM8K | MATH500 |
|---|---|---|
| LLaDA-8B-Instruct | 78.2 | 36.2 |
| + *diffu*-GRPO (Zhao et al., 2025) | 80.7 | 39.0 |
| + *d1* (Zhao et al., 2025) | 82.0 | 38.0 |
| + SDPO (Han et al., 2025) (full) | 81.2 | - |
| + TCR (Wang et al., 2025d) | 83.0 | 41.4 |
| + MDPO (He et al., 2025) (full) | 83.4 | 43.4 |
| + *wd1* | 82.3 | 39.0 |
| + *wd1* (full) | 82.7 | 43.6 |
| + *wd1++* (full) | **84.5** | **44.2** |

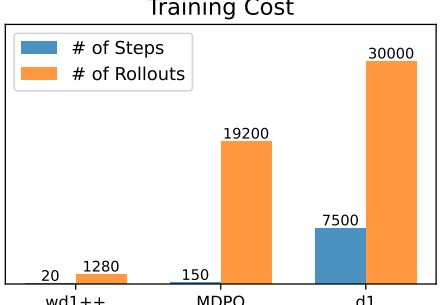

Table 4: Ablation on SFT and Negative Samples Weight ($w^-$). We conduct *wd1* training after SFT (*wd1*-SFT) and with only $w^+$ (namely *wd1*-P or WLL defined in Equation (6))[3]. Results show that *wd1* performs better without SFT on planning and math tasks. Removing negative sample reinforcement ($w^-$) significantly hurts performance, highlighting its importance.

| | Sudoku | | Countdown | | GSM8K | | MATH500 | |
|---|---|---|---|---|---|---|---|---|
| Model / Gen Len | 256 | 512 | 256 | 512 | 256 | 512 | 256 | 512 |
| *wd1*-P (WLL) | 6.69 | 6.84 | 13.67 | 4.69 | 65.66 | 78.17 | 29.40 | 22.80 |
| *wd1*-SFT | **26.5** | 24.2 | 43.4 | 43.4 | 80.7 | 82.0 | **36.4** | **39.0** |
| *wd1* | 25.2 | **24.2** | **51.2** | **46.1** | **80.8** | **82.3** | 34.4 | **39.0** |

We further assess sensitivity to the relative weighting of positive and negative samples. The combined weight (cw) corresponds to $\lambda$ in the mixture $-\lambda w^+ + (1-\lambda)w^-$, which scales the log-likelihood term in *wd1*. Training on negative samples alone (cw= 0.0) yields a pronounced deterioration in performance relative to our default setting (cw= 0.5). The results reinforce our argument that a balanced contribution of positive and negative weights is most effective. In the absence of positive samples, the reinforcement-learning signal collapses and optimisation becomes largely ineffective. A large emphasis on positive samples (cw= 0.8) causes performance to deteriorate more rapidly, highlighting the critical role of negative samples in weighted log-likelihood methods.

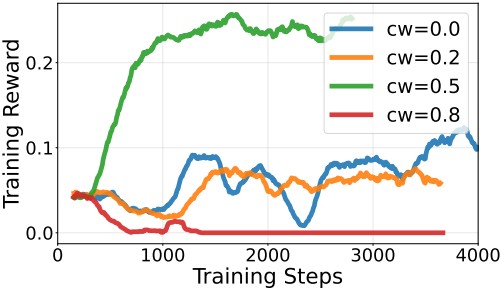

Figure 2: Training rewards of *wd1* under different combined weights on Sudoku.

# 6 RELATED WORK

**RL for Diffusion-based LLM.** RL for discrete diffusion models has been explored through several approaches. One line of work, exemplified by DRAKES (Wang et al., 2024), leverages reward back-propagation along the denoising trajectory. This approach requires computing a critic and propagating gradients through each denoising step, which is computationally intensive and prone to vanishing gradients. Alternatively, methods such as MMaDA (Yang et al., 2025) and *d1* (Zhao et al., 2025) adopt direct RL formulations like GRPO, approximating missing diffusion components—such as per-token likelihoods—for policy optimization. Zhu et al. (2025a) applies Direct Preference Optimization (DPO) to fine-tune the LLaDA base model (Nie et al., 2025a), achieving notable gains in reasoning tasks. However, these approaches all depend on likelihood ratios, which can introduce bias and instability due to likelihood approximation errors. In contrast, our method derives a weighted policy optimiza-

tion approach that eliminates the need for explicit policy ratios. Importantly, similar to prior works, our method directly optimizes the predictive distribution over clean data. A complementary line of research formulates policy optimization in terms of concrete scores (Lou et al., 2024; Meng et al., 2022). SEPO (Zekri & Boullé, 2025), for instance, introduces a policy optimization objective that only depends on concrete score estimation, thereby circumventing likelihood approximation altogether.

**RL for AR Models.** The connection between GRPO and weighted regression has recently been explored in the context of RL with verifier reward (Mroueh, 2025), where binary rewards simplify policy optimization into likelihood-based objectives. Other closely related approaches are Rejection Sampling Fine-Tuning (RAFT), which maximizes the likelihood of positive-reward samples (Xiong et al., 2025). Extensions of this idea incorporate negative samples to actively penalize the likelihood of negative-reward completions while enhancing that of high-reward ones (Zhu et al., 2025b; Chen et al., 2025). Other works introduce negative penalization through contrastive methods, such as Noise Contrastive Estimation (NCE) (Gutmann & Hyvärinen, 2012; van den Oord et al., 2019; Chen et al., 2024). Beyond binary rewards, preference-based learning has been widely studied using the Bradley–Terry model (Bradley & Terry, 1952; Ouyang et al., 2022; Rafailov et al., 2024; Azar et al., 2023; Ethayarajh et al., 2024; Wang et al., 2023; Hong et al., 2024). In contrast to these approaches, our method accommodates general reward signals and can be interpreted as a form of soft rejection sampling, enabling efficient and stable policy optimization for dLLMs.

**RL via Weighted Regression.** RL via weighted regression has been explored in earlier works advantage-weighted regression (AWR) (Peng et al., 2019; Peters et al., 2010), and more recently in the context of continuous control with diffusion policies (Ding et al., 2024; Zhang et al., 2025). Weighted likelihood-based approaches have also been proposed for fine-tuning autoregressive (AR) language models using general reward functions (Du et al., 2025; Baheti et al., 2024; Zhu et al., 2023). However, for AR models, where likelihoods are tractable, the necessity of such approaches remains unclear. In contrast, dLLMs suffer from intractable likelihoods, making weighted likelihood formulations particularly advantageous by reducing the number of required likelihood approximations. As such, RL via weighted likelihood provides a natural and efficient fit for optimizing dLLMs. In addition, we demonstrate in ablation study that merely optimizing policy with AWR (*wd1*-P) is ineffective.

**"Ratio-Free" Policy Optimization.** If a policy optimization objective requires neither importance sampling nor regularization with respect to a reference model, then the objective is ratio-free. Consequently, *on-policy* algorithms such as vanilla policy gradient methods (e.g., REINFORCE (Williams, 1992)) and their variants (e.g., RLOO (Kool et al., 2019)) are inherently ratio-free. This property is particularly valuable for dLLMs, where errors in log-likelihood approximation can accumulate and propagate through ratio-based computations. Concurrent work, such as SPG (Wang et al., 2025a), adopts a policy-gradient formulation and develops an objective tailored specifically for diffusion language models. Another *on-policy* optimization approach, d2 (Wang et al., 2025c), removes both the ratios and the likelihood terms from the RL objective for dLLMs, offering a more fundamental solution. However, our method *wd1*, similar to AWR (Peng et al., 2019), is inherently an *off-policy* loss, which is more general.

## 7   CONCLUSION

We introduce *wd1*, a weighted policy optimization method for reasoning with dLLMs. *wd1* is designed to minimize reliance on likelihood approximation, thereby mitigating the potentially substantial bias that can arise from approximation errors in policy ratios. Our method is grounded in a weighted log-likelihood objective, derived to approximate the closed-form solution to the reverse-KL-constrained policy optimization. Empirically, we show that *wd1*, even without supervised fine-tuning, surpasses the existing method *d1* by up to 16% in accuracy on reasoning benchmarks, while also delivering notable improvements in computational efficiency during RL training. These results highlight the effectiveness of *wd1* and establish it as a more scalable and efficient approach for fine-tuning dLLMs.

## 8   ETHICS AND REPRODUCIBILITY STATEMENT

This work raises no question or concern regarding the Code of Ethics. As for reproducibility of our results, we provide details of implementations in Section 5, in Experimental Setup and Implementation subsections. Additional details including dataset, reward functions, and hyperparameters are provided in Appendix B. All the theoretical results are proved in Appendix A.

## 9 ACKNOWLEDGMENTS

Ilija Bogunovic was supported by the EPSRC New Investigator Award EP/X03917X/1; the Engineering and Physical Sciences Research Council EP/S021566/1. Sangwoong Yoon was supported by the Institute of Information & Communications Technology Planning & Evaluation (IITP) grant funded by the Korea government (MSIT) (No. RS-2020-II201336, Artificial Intelligence Graduate School Program (UNIST)), the National Research Foundation of Korea(NRF) grant funded by the Korea government(MSIT) (No. RS-2024-00408003), and the Center for Advanced Computation at Korea Institute for Advanced Study. Xiaohang Tang was supported by the Engineering and Physical Sciences Research Council [grant number EP/T517793/1, EP/W524335/1]. Rares Dolga is supported by EPSRC, grant reference number EP/S021566/1.

The authors would like to thank UIPath and Che Liu (Imperial College London) for providing computing resources that supported our experiments, and Prof. David Barber, Yiming Yang, Xiaoyuan Cheng, and Keyue Jiang from University College London for valuable discussions during the early stages of this work.

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

CONTENTS

# A  PROOFS AND ADDITIONAL THEORY

## A.1  OBJECTIVE ESTIMATION ERROR DUE TO LIKELIHOOD APPROXIMATION

In this section, we aim to show that *diffu*-GRPO amplify the log-likelihood approximation error. Denote the approximator by $\phi$ such that $\|\phi^{\pi_\theta}(q, o) - \log \pi_\theta(o|q)\| \leq \epsilon$ and $\|\phi^{\pi_{\text{old}}}(q, o) - \log \pi_{\text{old}}(o|q)\| \leq \epsilon'$. Then the objective *diffu*-GRPO in the worst case suffers from exponential error. We discuss the case without ratio clipping and omit the regularization for convenience. Denote $\mathcal{L}_{\text{GRPO}}$ as the ground truth objective without likelihood approximation:

$$\|\mathcal{L}_{\textit{diffu}\text{-GRPO}} - \mathcal{L}_{\text{GRPO}}\|$$

$$= \Big\| \mathbb{E}_{q \sim \mathcal{D}, \, o_{1:G} \sim \pi_{\text{old}}(\cdot|q)} \Big[ \frac{1}{G} \sum_{i=1}^{G} \frac{1}{|o_i|} \sum_{k=1}^{|o_i|} \big( \exp \phi^{\pi_\theta}(o_i^k) / \exp \phi^{\pi_{\text{old}}}(o_i^k) \big) \hat{A}_i \Big]$$

$$- \mathbb{E}_{q \sim \mathcal{D}, \, o_{1:G} \sim \pi_{\text{old}}(\cdot|q)} \Big[ \frac{1}{G} \sum_{i=1}^{G} \frac{1}{|o_i|} \sum_{k=1}^{|o_i|} \big( \pi_\theta(o_i^k) / \pi_{\text{old}}(o_i^k) \big) \hat{A}_i \Big] \Big\|$$

$$= \Big\| \mathbb{E}_{q \sim \mathcal{D}, \, o_{1:G} \sim \pi_{\text{old}}(\cdot|q)} \Big[ \frac{1}{G} \sum_{i=1}^{G} \frac{1}{|o_i|} \sum_{k=1}^{|o_i|} \exp \big( \phi^{\pi_\theta}(o_i^k) - \phi^{\pi_{\text{old}}}(o_i^k) \big) \hat{A}_i \Big]$$

$$- \mathbb{E}_{q \sim \mathcal{D}, \, o_{1:G} \sim \pi_{\text{old}}(\cdot|q)} \Big[ \frac{1}{G} \sum_{i=1}^{G} \frac{1}{|o_i|} \sum_{k=1}^{|o_i|} \big( \pi_\theta(o_i^k) / \pi_{\text{old}}(o_i^k) \big) \hat{A}_i \Big] \Big\|$$

$$\leq \Big\| \mathbb{E}_{q \sim \mathcal{D}, \, o_{1:G} \sim \pi_{\text{old}}(\cdot|q)} \Big[ \frac{1}{G} \sum_{i=1}^{G} \frac{1}{|o_i|} \sum_{k=1}^{|o_i|} \exp \big( \log \pi_\theta(o_i^k) - \log \pi_{\text{old}}(o_i^k) + (\epsilon + \epsilon') \big) \hat{A}_i \Big]$$

$$- \mathbb{E}_{q \sim \mathcal{D}, \, o_{1:G} \sim \pi_{\text{old}}(\cdot|q)} \Big[ \frac{1}{G} \sum_{i=1}^{G} \frac{1}{|o_i|} \sum_{k=1}^{|o_i|} \big( \pi_\theta(o_i^k) / \pi_{\text{old}}(o_i^k) \big) \hat{A}_i \Big] \Big\|$$

$$= \Big\| \mathbb{E}_{q \sim \mathcal{D}, \, o_{1:G} \sim \pi_{\text{old}}(\cdot|q)} \Big[ \frac{1}{G} \sum_{i=1}^{G} \frac{1}{|o_i|} \sum_{k=1}^{|o_i|} \exp \big( \epsilon + \epsilon' \big) \hat{A}_i \Big] \Big\| \leq C \exp \big( \epsilon + \epsilon' \big), \tag{15}$$

where $C$ is a constant independent to $\epsilon$ and $\epsilon'$. In contrast *wd1* has only linear approximation error. Denote the objective computed using approximated log-likelihood as $\mathcal{L}_\phi$

$$\|\mathcal{L}_\phi - \mathcal{L}_{\textit{wd1}}\| = \Big\| \mathbb{E}_{q \sim \mathcal{D}, \{o_i\}_{i=1}^G \sim \pi_{\text{old}}^{\text{ref}}(\cdot|q)} \Big[ \sum_{i=1}^{G} \big( - w^+(q, o_i) + w^-(q, o_i) \big) \cdot \big( \phi(q, o_i) - \log \pi_\theta(o_i|q) \big) \Big] \Big\|$$

$$\leq C'\epsilon. \tag{16}$$

## A.2  REINFORCEMENT LEARNING

**Reinforcement Learning Formulation.** We first introduce the reinforcement learning notations and then extend it to the setting of LLM post-training. Denote $\tau$ as a trajectory ($\tau = (s_0, a_0, s_1, \dots) \sim \pi$) sampled following policy $\pi$. Specifically, $s_0 \sim \mu$, $a_t \sim \pi(\cdot|s_t)$, $s_{t+1} \sim P(\cdot|q_t, a_t)$. The objective of Reinforcement Learning aims to find policy $\pi$, which maximizes a discounted total return,

$$\eta(\pi) = \mathbb{E}_{\tau \sim \pi} \Big[ \sum_{t=0}^{\infty} \gamma^t r(s_t, a_t, s_{t+1}) \Big].$$

Let the discounted return of a trajectory be $R(\tau) = \sum_{t=0}^{\infty} \gamma^t r(s_t, a_t, s_{t+1})$. The advantage function is defined as $A^\pi(s, a) = Q^\pi(s, a) - V^\pi(s)$, where $V^\pi(s) = \mathbb{E}_{\tau \sim \pi}[R(\tau)|s_0 = s]$ is state value function, and $Q^\pi(s, a) = \mathbb{E}_{\tau \sim \pi}[R(\tau)|s_0 = s, a_0 = a]$ is state-action value function. Denote $\rho_{\pi_{\text{old}}}$ as the marginal state distribution. Denote the total variation of two discrete probability distributions $a, b$ by $D_{TV}(a, b) := \frac{1}{2} \sum_i |a_i - b_i|$ and $D_{TV}(a, b)^2 \leq D_{\text{KL}}(a \parallel b)$ (Pollard, 2000; Schulman et al., 2015). When $a$ and $b$ are conditional probability distribution, denote $D_{TV}^{\max}(a, b) = \max_q D_{TV}(a(\cdot|q) \parallel b(\cdot|q))$ and $D_{\text{KL}}^{\max}(a \parallel b) = \max_q D_{\text{KL}}(a(\cdot|q) \parallel b(\cdot|q))$.

We then extend RL for LLM post-training. In this paper we only consider the sequence-level reward and loss objective, so we directly replace $s$ with $q$ and $a$ with completion $o$. Then the horizon of the RL for post-training becomes only 1. The following theorem provides a monotonic (non-decreasing) guarantee of existing prevailing RL methods.

**Proposition 1** (Policy Improvement Bound (Kakade & Langford, 2002; Schulman et al., 2015)). *Let surrogate objective $L_{\pi_{old}}(\pi) = \eta(\pi_{old}) + \mathbb{E}_{s \sim \rho_{\pi_{old}}(\cdot),\, a \sim \pi(\cdot|s)}\big[A^{\pi_{old}}(s,a)\big]$, and $C = 4 \max_{s,a,\pi} |A^{\pi}(s,a)|\gamma/(1-\gamma)^2$, then $\forall k \in \mathbb{N}$:*

$$\eta(\pi^*) \geq L_{\pi_{old}}(\pi^*) - C D_{TV}^{\max}(\pi_{old}, \pi^*)^2.$$

**Remark 3.** *Based on Proposition 1, due to $D_{TV}^{\max}(a||b)^2 \leq D_{KL}^{\max}(a||b)$ (Pollard, 2000; Schulman et al., 2015), TRPO and PPO with fixed forward KL regularization have the monotonic improvement guarantees. In other words, $\eta(\pi^*) \geq L_{\pi_{old}}(\pi^*) - C D_{TV}^{\max}(\pi_{old}, \pi^*)^2 \geq L_{\pi_{old}}(\pi^*) - C\mathbb{E}[D_{KL}(\pi_{old}||\pi^*)] \geq L_{\pi_{old}}(\pi_{old}) = \eta(\pi_{old})$.*

**Proposition 2.** *Minimizing $D_{KL}(\pi^*(\cdot|q) \| \pi_\theta(\cdot|q))$ w.r.t. $\theta$ is equivalent to optimize the following loss objective:*

$$\mathcal{L}_{WLL}(\theta) = \mathbb{E}_{q \sim \mathcal{D}, o \sim \pi_{old}^{ref}(\cdot|q)}\Big[ -\exp\big(\psi A^{\pi_{old}}(q,o)\big) \cdot \log \pi_\theta(o_i|q)\Big] \tag{17}$$

$$\approx -\mathbb{E}_{\{o_i\}_{i=1}^G \sim \pi_{old}^{ref}(o|q)}\Big[\frac{1}{G}\sum_{i=1}^G \frac{\exp\big(\psi A^{\pi_{old}}(q,o_i)\big)}{\sum_{j=1}^G[\exp\big(\psi A^{\pi_{old}}(q,o_j)\big)]} \cdot \log \pi_\theta(o_i|q)\Big]. \tag{18}$$

*Proof.* To obtain the practical objective in Equation (18), we first start from the cross-entropy loss, and obtain the following. $\forall q \in \mathcal{D}$:

$$D_{KL}(\pi^*(\cdot|q) \| \pi_\theta(\cdot|q))$$

$$= -\mathbb{E}_{o \sim \pi^*(\cdot|q)}\Big[\log \pi_\theta(o|q)\Big] \tag{19}$$

$$= -\Big[\sum_o \pi^*(o|q) \cdot \log \pi_\theta(o|q)\Big] \tag{20}$$

$$= -\Big[\sum_o \frac{\pi_{old}^{ref}(o|q)\exp\big(\psi A^{\pi_{old}}(q,o)\big)}{\sum_{o'}\pi_{old}^{ref}(o'|q)[\exp\big(\psi A^{\pi_{old}}(q,o')\big)]} \cdot \log \pi_\theta(o|q)\Big] \tag{21}$$

$$= -\mathbb{E}_{o \sim \pi_{old}^{ref}(o|q)}\Big[\frac{\exp\big(\psi A^{\pi_{old}}(q,o)\big)}{\mathbb{E}_{o' \sim \pi_{old}^{ref}}[\exp\big(\psi A^{\pi_{old}}(q,o')\big)]} \cdot \log \pi_\theta(o|q)\Big] \tag{22}$$

Since the normalization constant $\mathbb{E}_{o' \sim \pi_{old}^{ref}}[\exp\big(\psi A^{\pi_{old}}(q,o')\big)]$ is independent to $o$, we can convert the objective to a weighted log-likelihood, and approximate it with samples from the group and weight normalization to obtain:

$$\mathcal{L}_{WLL}(\theta) = -\mathbb{E}_{o \sim \pi_{old}^{ref}(o|q)}\Big[\exp\big(\psi A^{\pi_{old}}(q,o)\big) \cdot \log \pi_\theta(o|q)\Big] \tag{23}$$

$$\approx -\mathbb{E}_{\{o_i\}_{i=1}^G \sim \pi_{old}^{ref}(o|q)}\Big[\frac{1}{G}\sum_{i=1}^G \frac{\exp\big(\psi A^{\pi_{old}}(q,o_i)\big)}{\sum_{j=1}^G[\exp\big(\psi A^{\pi_{old}}(q,o_j)\big)]} \cdot \log \pi_\theta(o_i|q)\Big]. \tag{24}$$

We derive Equation (21) from Equation (20) by simply using the known form of the optimal policy $\pi^*(\cdot|q) \propto \pi_{old}^{ref}(\cdot|q) \cdot \exp\big(\psi \hat{A}(q,\cdot)\big)$. We derive Equation (22) from Equation (21) by using the definition of expectation and from Equation (22) to Equation (24) by approximating through $G$ samples $\{o_i\}_{i=1}^G \sim \pi_{old}^{ref}(o|q)$.

$\square$

**Theorem 2.** *Reverse-KL-regularized Policy Optimization defined in the following objective has monotonic improvement guarantees. Specifically, denote regularized objective $\eta'(\pi) = \eta(\pi) - \mathbb{E}_{q \in \mathcal{D}}\big[\beta D_{KL}\big(\pi(\cdot|q) \| \pi_{ref}(\cdot|q)\big)\big]$ and denote*

$$M(\pi) = L(\pi) - \mathbb{E}_{q \in \mathcal{D}}\Big[\lambda D_{KL}(\pi(\cdot|q)\|\pi_{old}(\cdot|q)) + \beta D_{KL}\big(\pi(\cdot|q) \| \pi_{ref}(\cdot|q)\big)\Big], \tag{25}$$

where $L(\pi) = \eta(\pi_{old}) + \mathbb{E}_{q\sim\mathcal{D},\ o\sim\pi(\cdot|q)}\big[A^{\pi_{old}}(q,o)\big]$. Let $\theta^*$ be the solution to the objective $\max_\theta M(\pi_\theta)$:

$$\theta^* = \arg\max_\theta\ \mathbb{E}_{q\in\mathcal{D},o\sim\pi_\theta(\cdot|q)}\Big[A^{\pi_{old}}(q,o) - \lambda D_{KL}\big(\ \pi_\theta(\cdot|q)\ \|\ \pi_{old}(\cdot|q)\ \big) - \beta D_{KL}\Big(\pi_\theta(\cdot|q)\ \|\ \pi_{ref}(\cdot|q)\Big)\Big] \tag{26}$$

then $\eta'(\pi^*) \geq \eta'(\pi_{old})$.

*Proof.* Based on Proposition 1, we have

$$\eta'(\pi^*) = \eta(\pi^*) - \mathbb{E}_{q\in\mathcal{D}}\Big[\beta D_{KL}\Big(\pi^*(\cdot|q)\ \|\ \pi_{ref}(\cdot|q)\Big)\Big]$$

$$\geq L(\pi^*) - CD_{TV}^{\max}(\pi_{old},\pi^*)^2 - \mathbb{E}_{q\in\mathcal{D}}\Big[\beta D_{KL}\Big(\pi^*(\cdot|q)\ \|\ \pi_{ref}(\cdot|q)\Big)\Big] \tag{27}$$

$$\geq L(\pi^*) - CD_{KL}^{\max}(\pi^*\|\pi_{old}) - \mathbb{E}_{q\in\mathcal{D}}\Big[\beta D_{KL}\Big(\pi^*(\cdot|q)\ \|\ \pi_{ref}(\cdot|q)\Big)\Big] \tag{28}$$

$$\geq L(\pi^*) - \mathbb{E}_{q\in\mathcal{D}}\Big[\lambda D_{KL}(\pi^*(\cdot|q)\|\pi_{old}(\cdot|q)) + \beta D_{KL}\Big(\pi^*(\cdot|q)\ \|\ \pi_{ref}(\cdot|q)\Big)\Big] \tag{29}$$

$$= M(\pi^*) \tag{30}$$

$$\geq M(\pi_{old}) \tag{31}$$

$$= L(\pi_{old}) - \mathbb{E}_{q\in\mathcal{D}}\Big[\lambda D_{KL}(\pi_{old}(\cdot|q)\|\pi_{old}(\cdot|q)) + \beta D_{KL}\Big(\pi_{old}(\cdot|q)\ \|\ \pi_{ref}(\cdot|q)\Big)\Big] \tag{32}$$

$$\geq L(\pi_{old}) - \mathbb{E}_{q\in\mathcal{D}}\Big[\beta D_{KL}\Big(\pi_{old}(\cdot|q)\ \|\ \pi_{ref}(\cdot|q)\Big)\Big] \tag{33}$$

$$= \eta(\pi_{old}) - \mathbb{E}_{q\in\mathcal{D}}\Big[\beta D_{KL}\Big(\pi_{old}(\cdot|q)\ \|\ \pi_{ref}(\cdot|q)\Big)\Big] \tag{34}$$

$$= \eta'(\pi_{old}) \tag{35}$$

Equation (27) holds due to Proposition 1. Equation (28) holds due to $D_{TV}^{\max}(p\|q)^2 \leq D_{KL}^{\max}(p\|q)$ (Pollard, 2000). Equation (29) holds due to the definition of $D_{KL}^{\max}$. Equation (30) is according to the definition of $M(\cdot)$. The key inequality Equation (31) holds since $\pi^*$ is the maximizer of function $L(\cdot)$. Equation (32) holds due to the definition of $M(\cdot)$. Equation (33) holds since $D_{KL}(\pi_{old}(\cdot|q)\|\pi_{old}(\cdot|q)) = 0$. Equation (34) holds since $L(\pi_{old}) = \eta(\pi_{old}) + \mathbb{E}_{q\sim\mathcal{D},\ o\sim\pi_{old}(\cdot|q)}\big[A^{\pi_{old}}(q,o)\big] = \eta(\pi_{old})$. Equation (35) is from the definition of $\eta'$. $\qquad\square$

### A.3 MASKED DISCRETE DIFFUSION

In this section, we show how our objective learns a distribution for which all marginals at time $t$ satisfy intermediate energy guidance as per Lu et al. (2023).

**Definition 2.** *The absorbing transition kernel is defined as $Q_t = \sigma(t)Q^{absorb}$, where*

$$Q^{absorb} = \begin{bmatrix} -1 & 0 & \cdots & 0 & 1 \\ 0 & -1 & \cdots & 0 & 1 \\ \vdots & \vdots & \ddots & \vdots & \vdots \\ 0 & 0 & \cdots & -1 & 1 \\ 0 & 0 & \cdots & 0 & 0 \end{bmatrix}.$$

**Definition 3** (Concrete Score). *Denote $x_t = (x_t^1, \cdots, x_t^d)$ and $\hat{x}_t$ is identical to $x_t$ except the $i$-th token is unmasked (i.e. $x_t^i = [M]$ and $\hat{x}_t^i \neq [M]$). Concrete score is defined as the marginal probability ratio between $\hat{x}_t$ and $x_t$:*

$$s(x_t, t) \stackrel{def}{=} \frac{p(x_t^1, \cdots, \hat{x}_t^i, \cdots, x_t^d)}{p(x_t^1, \cdots, x_t^i, \cdots, x_t^d)}. \tag{36}$$

**Proposition 3** (**Marginal Distribution** (Ou et al., 2025b)). *Denote $\{x_t\}$ as a continuous time Markov chain with transition matrix $\mathbf{Q}_t = \sigma(t)\mathbf{Q}^{absorb}$. Assume $d_1$ tokens in $x_t = (x_t^1, \cdots, x_t^d)$ are masked tokens $[\mathbf{M}]$, and $d_2 = d - d_1$ tokens are unmasked, the marginal distribution $p_t(x_t)$ satisfies*

$$p_t(x_t) = \big[1 - e^{-\bar{\sigma}(t)}\big]^{d_1}\big[e^{-\bar{\sigma}(t)}\big]^{d_2} p_0(x_t^{UM}), \tag{37}$$

*where $\bar{\sigma}(t) = \int_0^t \sigma(s)ds$, and $x_t^{UM}$ is the set of unmasked tokens in $x_t$.*

The following theorem provides the foundation of directly modeling the clean data distribution.

**Proposition 4** (**Analytic Concrete Score** (Ou et al., 2025b)). *Denote $x_t = (x_t^1, \cdots, x_t^d)$ and $\hat{x}_t$ is identical to $x_t$ except the $i$-th token is unmasked (i.e. $x_t^i = [M]$ and $\hat{x}_t^i \neq [M]$). Then the concrete score at time $t$ can be expressed by the conditional probability of predicting this unmasked token.*

$$\frac{p_t(x_t^1 \ldots \hat{x}_t^i \ldots x_t^d)}{p_t(x_t^1 \ldots x_t^i \ldots x_t^d)} = \frac{e^{-\bar{\sigma}(t)}}{1 - e^{-\bar{\sigma}(t)}} p_0(\hat{x}_t^i \mid x_t^{UM})$$

.

**Lemma** (**1**). *The marginal probability distribution of the masked responses ($x_t$) in the diffusion process satisfies $p_t^*(x_t) = p_t'(x_t) \cdot \exp\big(A_t(x_t)\big)/Z_t$, which induces an energy-guided discrete diffusion:*

$$p_{0|t}^*(x_0|x_t) \propto p_{0|t}'(x_0|x_t) \cdot \exp(A(x_0) - A_t(x_t)), \tag{38}$$

*where intermediate energy function is defined as $A_t(x_0) = \log \mathbb{E}_{x_0 \sim p_{0|t}'(\cdot|x_t)}[\exp\big(A(x_0)\big)]$ for $t > 0$, and $A_0(x_0) = A(x_0)$, $A(\cdot)$ is advantage function, $Z_t$ is the normalization constant.*

*Proof.* The theorem and proof mainly extend from theory developed in continuous setting (Lu et al., 2023). According to the marginal likelihood of clean data distribution $p_0^*(x_0) = p_0'(x_0)\frac{e^{A(x_0)}}{Z}$, and identical forward process, we can rewrite the marginal likelihood of masked data:

$$p_t^*(x_t) = \int p_{t|0}^*(x_t|x_0)p_0^*(x_0)\,\mathrm{d}x_0 = \int p_{t|0}^*(x_t|x_0)p_0'(x_0)\frac{e^{\psi A(x_0)}}{Z}\,\mathrm{d}x_0$$

$$= \int p_{t|0}'(x_t|x_0)p_0'(x_0)\frac{e^{\psi A(x_0)}}{Z}\,\mathrm{d}x_0.$$

Applying Bayesian rule we know that $p_{t|0}'(x_t|x_0)p_0'(x_0) = p_{0|t}'(x_0|x_t)p_t'(x_t)$, hence we can further rewrite

$$p_t^*(x_t) = \int p_{t|0}'(x_t|x_0)p_0'(x_0)\frac{e^{\psi A(x_0)}}{Z}\,\mathrm{d}x_0 = p_t'(x_t)\int p_{0|t}'(x_0|x_t)\frac{e^{\psi A(x_0)}}{Z}\,\mathrm{d}x_0$$

$$= \frac{p_t'(x_t)\,\mathbb{E}_{p_{0|t}'(x_0|x_t)}\big[e^{\psi A(x_0)}\big]}{Z} = \frac{p_t'(x_t)\,e^{\psi A_t(x_t)}}{Z_t}$$

Therefore, the marginal likelihood of masked sequence satisfies: $p_t^*(x_t) = p_t'(x_t) \cdot \exp\big(A_t(x_t)\big)/Z_t$. Since $p_{t|0}^* = p_{t|0}'$, based on the marginal likelihood of clean data distribution satisfies $p_0^*(x_0) = p_0'(x_0)\frac{e^{A(x_0)}}{Z}$, we can further applying Bayesian rule to obtain the energy-guided discrete diffusion model:

$$p_{0|t}^*(x_0|x_t) = \frac{p_{t|0}^*(x_t|x_0)p_0^*(x_0)}{p_t^*(x_t)} \tag{39}$$

$$= \frac{p_{t|0}'(x_t|x_0)p_0'(x_0)\frac{e^{A(x_0)}}{Z}}{p_t^*(x_t)} \tag{40}$$

$$= \frac{p_{t|0}'(x_t|x_0)p_0'(x_0)\frac{e^{A(x_0)}}{Z}}{\frac{p_t'(x_t)\,e^{\psi A_t(x_t)}}{Z_t}} \tag{41}$$

$$\propto p_{0|t}'(x_0|x_t) \cdot \exp(A(x_0) - A_t(x_t)), \tag{42}$$

$\square$

**Lemma 2.** *According to Definition 1, due to the identical forward process*

$$p_{t|0}^*(x_t|x_0) = p_{t|0}'(x_t|x_0) = p_{t|0}^{ref}(x_t|x_0), \tag{43}$$

*based on Lemma 1 and Proposition 3, we have the marginal probability of the unmasked tokens satisfies that for all step t,*

$$p_0^*(x_t^{UM}|q) = p_0(x_t^{UM}|q)^\lambda \cdot p_0^{ref}(x_t^{UM}|q)^\beta \cdot \mathbb{E}_{p_0'(x_0|x_t)}[\exp\big(A(q, x_0)\big)]/Z, \tag{44}$$

*where Z is the normalization constant.*

*Proof.* According to the identical forward distribution of three diffusion process (new, old, and reference), based on Equation (37), we have $\forall t$:

$$p_t(x_t|q) = \left[1 - e^{-\bar{\sigma}(t)}\right]^{d_1} \left[e^{-\bar{\sigma}(t)}\right]^{d_2} p_0(x_t^{\text{UM}}|q) \tag{45}$$

$$p_t^*(x_t|q) = \left[1 - e^{-\bar{\sigma}(t)}\right]^{d_1} \left[e^{-\bar{\sigma}(t)}\right]^{d_2} p_0^*(x_t^{\text{UM}}|q) \tag{46}$$

$$p_t^{\text{ref}}(x_t|q) = \left[1 - e^{-\bar{\sigma}(t)}\right]^{d_1} \left[e^{-\bar{\sigma}(t)}\right]^{d_2} p_0^{\text{ref}}(x_t^{\text{UM}}|q) \tag{47}$$

Then rewrite Equation (46) in the residual energy-based form defined in Equation (38), we have

$$\left[1 - e^{-\bar{\sigma}(t)}\right]^{d_1} \left[e^{-\bar{\sigma}(t)}\right]^{d_2} p_0^*(x_t^{\text{UM}}|q) = p_t^*(x_t|q) = p_t'(x_t|q) \cdot \exp\left(A_t(q, x_t)\right)/Z. \tag{48}$$

By plugging $p_t'(x_t|q) = p_t(x_t|q)^\lambda \cdot p_t^{\text{ref}}(x_t|q)^\beta$ and Equation (45) and Equation (47) into Equation (48), we have that the clean data distribution of the unmask tokens at diffusion time $t$ satisfies:

$$p_0^*(x_t^{\text{UM}}|q) = p_0(x_t^{\text{UM}}|q)^\lambda \cdot p_0^{\text{ref}}(x_t^{\text{UM}}|q)^\beta \cdot \exp\left(A_t(q, x_t)\right)/Z \tag{49}$$

$$= p_0(x_t^{\text{UM}}|q)^\lambda \cdot p_0^{\text{ref}}(x_t^{\text{UM}}|q)^\beta \cdot \mathbb{E}_{p_0'(x_0|x_t)}[\exp\left(A(q, x_0)\right)]/Z. \tag{50}$$

$\square$

**Proposition 5.** *The marginal likelihood of the target diffusion model $p^*$ satisfies Equation (38). Consequently, the concrete score of the target diffusion model, denoted by $s^*$, can be expressed by the score of the mixture diffusion $p'$ and the posterior mean of the advantage:*

$$s^*(x_t, t) = s'(x_t, t) \cdot \frac{\mathbb{E}_{p_0'(x_0|\hat{x}_t)}[\exp\left(A(x_0)\right)]/\hat{Z}}{\mathbb{E}_{p_0'(x_0|x_t)}[\exp\left(A(x_0)\right)]/Z}, \tag{51}$$

*and equivalently*

$$p_0(\hat{x}_t^i|x_t^{UM}, q)^\lambda \cdot p_0^{ref}(\hat{x}_t^i|x_t^{UM}, q)^\beta \cdot \frac{\mathbb{E}_{p_0'(x_0|\hat{x}_t)}[\exp\left(A(q, x_0)\right)]/\hat{Z}}{\mathbb{E}_{p_0'(x_0|x_t)}[\exp\left(A(q, x_0)\right)]/Z}. \tag{52}$$

*Proof.* According to Lemma 2

$$\frac{p_0^*(x_t^{\text{UM}}, \hat{x}_t^i|q)}{p_0^*(x_t^{\text{UM}}|q)} = \frac{p_0(x_t^{\text{UM}}, \hat{x}_t^i|q)^\lambda}{p_0(x_t^{\text{UM}}|q)^\lambda} \cdot \frac{p_0^{\text{ref}}(x_t^{\text{UM}}, \hat{x}_t^i|q)^\beta}{p_0^{\text{ref}}(x_t^{\text{UM}}|q)^\beta} \cdot \frac{\mathbb{E}_{p_0'(x_0|\hat{x}_t)}[\exp\left(A(q, x_0)\right)]/\hat{Z}}{\mathbb{E}_{p_0'(x_0|x_t)}[\exp\left(A(q, x_0)\right)]/Z} \tag{53}$$

$$p_0^*(\hat{x}_t^i|x_t^{\text{UM}}, q) = p_0(\hat{x}_t^i|x_t^{\text{UM}}, q)^\lambda \cdot p_0^{\text{ref}}(\hat{x}_t^i|x_t^{\text{UM}}, q)^\beta \cdot \frac{\mathbb{E}_{p_0'(x_0|\hat{x}_t)}[\exp\left(A(q, x_0)\right)]/\hat{Z}}{\mathbb{E}_{p_0'(x_0|x_t)}[\exp\left(A(q, x_0)\right)]/Z}. \tag{54}$$

Both sides in Equation (52) multiply $C(t) = \frac{e^{-\cdot\bar{\sigma}(t)}}{1-e^{\bar{\sigma}(t)}}$ and based on the analytic form of concrete score introduced in Proposition 4, $C(t) \cdot p_0(\hat{x}_t^i|x_t^{\text{UM}}, q) = s(x_t^i, t)$. Thus, we have

$$C(t) \cdot p_0^*(\hat{x}_t^i|x_t^{\text{UM}}, q) = C(t) \cdot p_0(\hat{x}_t^i|x_t^{\text{UM}}, q)^\lambda \cdot p_0^{\text{ref}}(\hat{x}_t^i|x_t^{\text{UM}}, q)^\beta \cdot \frac{\mathbb{E}_{p_0'(x_0|\hat{x}_t)}[\exp\left(A(q, x_0)\right)]/\hat{Z}}{\mathbb{E}_{p_0'(x_0|x_t)}[\exp\left(A(q, x_0)\right)]/Z} \tag{55}$$

$$s^*(x_t, t) = s'(x_t, t) \cdot \frac{\mathbb{E}_{p_0'(x_0|\hat{x}_t)}[\exp\left(A(q, x_0)\right)]/\hat{Z}}{\mathbb{E}_{p_0'(x_0|x_t)}[\exp\left(A(q, x_0)\right)]/Z}. \tag{56}$$

$\square$

**Lemma 3.** *The normalization constant $Z = \sum_{x_t} p'(x_t|q) \cdot \mathbb{E}_{x_0|x_t}[\exp A(q, x_0)]$ is independent to the masked response $x_t$. In other words, $Z = \hat{Z} := \sum_{\hat{x}_t} p'(\hat{x}_t|q) \cdot \mathbb{E}_{x_0|\hat{x}_t}[\exp A(q, x_0)]$ for any $\hat{x}_t \neq x_t$.*

*Proof.*

$$Z = \sum_{x_t} p'(x_t|q) \cdot \mathbb{E}_{x_0|x_t}[\exp A(q,x_0)] = \sum_{x_t} p'(x_t|q) \cdot \sum_{x_0} p'(x_0|x_t) \cdot \exp A(q,x_0) \quad (57)$$

$$= \sum_{x_t} \sum_{x_0} p'(x_0,x_t|q) \cdot \exp A(q,x_0) = \sum_{x_0} \sum_{x_t} p'(x_0,x_t|q) \cdot \exp A(q,x_0) \quad (58)$$

$$= \sum_{x_0} p'(x_0|q) \cdot \exp A(q,x_0) \quad (59)$$

Thus $Z$ becomes independent to $x_t$, leading to that

$$Z = \hat{Z} := \sum_{\hat{x}_t} p'(\hat{x}_t|q) \cdot \mathbb{E}_{x_0|\hat{x}_t}[\exp A(q,x_0)] \quad (60)$$

$\square$

**Theorem (1).** *The score model $s_\theta = s^*$ defined in Equation (52) is satisfied when the following loss objective is minimized. This objective is in a form of **advantage-weighted** Denoising Concrete Score Matching (D-CSM), which we call AW-D-CSM:*

$$\mathcal{L}_{\text{AW-D-CSM}} = \mathbb{E}_{p_0'(x_0)}[\underbrace{\exp\left(A(q,x_0)\right)}_{\text{Advantage Weight}} \cdot \underbrace{\mathbb{E}_{t\sim[0,T],p_{t|0}'(x_t|x_0)}[\|s_\theta(x_t,t) - \frac{p_0'(\hat{x}_t|x_0)}{p_0'(x_t|x_0)}\|_2^2]}_{\mathcal{L}_{\text{D-CSM}}(x_0)}]. \quad (61)$$

*Proof.* Denote $s_\theta(x_t,t) = \frac{e^{-\bar{\sigma}(t)}}{1-e^{-\bar{\sigma}(t)}} p_\theta(\hat{x}_t^i \mid x_t^{\text{UM}})$ is the concrete score model induced by $p_\theta$. According to Lemma 3, $\hat{Z} = Z$. Then according to Proposition 5, Equation (52) is equivalent to

$$p_0^*(\hat{x}_t^i|x_t^{\text{UM}},q) \cdot \mathbb{E}_{p_0'(x_0|x_t)}[\exp\left(A(q,x_0)\right)]$$
$$= p_0(\hat{x}_t^i|x_t^{\text{UM}},q)^\lambda \cdot p_0^{\text{ref}}(\hat{x}_t^i|x_t^{\text{UM}},q)^\beta \cdot \mathbb{E}_{p_0'(x_0|\hat{x}_t)}[\exp\left(A(q,x_0)\right)] \quad (62)$$

We aim to update $p_\theta(\hat{x}_t^i|x_t^{\text{UM}},q) \to p_0^*(\hat{x}_t^i|x_t^{\text{UM}},q)$ to satisfy Equation (62), thus we can construct a loss function objective by replacing $p^*$ with $p_\theta$ and construct a $L^2$ norm loss

$$\mathbb{E}_{p_0'(x_0|x_t)}[\exp\left(A(q,x_0)\right) \cdot \|p_\theta(\hat{x}_t^i|x_t^{\text{UM}},q) - \frac{p_0'(x_0|\hat{x}_t)}{p_0'(x_0|x_t)} \cdot p_0(\hat{x}_t^i|x_t^{\text{UM}},q)^\lambda p_0^{\text{ref}}(\hat{x}_t^i|x_t^{\text{UM}},q)^\beta\|_2^2]$$
$$(63)$$

$$= \mathbb{E}_{p_0'(x_0|x_t)}[\exp\left(A(q,x_0)\right) \cdot \|p_\theta(\hat{x}_t^i|x_t^{\text{UM}},q) - \frac{p_0'(x_0|\hat{x}_t)}{p_0'(x_0|x_t)} \cdot p_0'(\hat{x}_t^i|x_t^{\text{UM}},q)\|_2^2] \quad (64)$$

$$= \mathbb{E}_{p_0'(x_0|x_t)}[\exp\left(A(q,x_0)\right) \cdot \|p_\theta(\hat{x}_t^i|x_t^{\text{UM}},q) - \frac{p_0'(\hat{x}_t|x_0)p_t'(x_t)}{p_0'(x_t|x_0)p_t'(\hat{x}_t)} \cdot p_0'(\hat{x}_t^i|x_t^{\text{UM}},q)\|_2^2] \quad (65)$$

$$= \mathbb{E}_{p_0'(x_0|x_t)}[\exp\left(A(q,x_0)\right) \cdot \|s_\theta(x_t,t) - \frac{p_0'(\hat{x}_t|x_0)}{p_0'(x_t|x_0)}\|_2^2]. \quad (66)$$

$\square$

# B ADDITIONAL EXPERIMENT SETUP DETAILS

## B.1 DATASET, TRAINING AND EVALUATION PROTOCOL

As for *wd1* and *d1*, we reproduce *d1* by running the official code[4] without and change, and train our method *wd1* evaluated for accuracy of the test datasets at steps 1000, 2500, 5000, 7500 in both GSM8k and MATH; at steps 1000, 2500, 4000, 5000, 12500 in Sudoku; and at 1000, 2500, 4000 in Countdown. We evaluate less checkpoints compared to *d1*. On the GSM8K, we train models on

---

[4]https://github.com/dllm-reasoning/d1

the train split[5] and evaluate on the test split. On Countdown, we train on the 3-number subset of the dataset[6] from TinyZero (Pan et al., 2025), and evaluate on 256 synthetic 3-number questions provided by Zhao et al. (2025). On Sudoku we use the 4×4 dataset[7] generated by Arel (2025). We train on 1M unique puzzles and evaluate on 256 synthetic ones provided by Zhao et al. (2025). On MATH500, we train models on the train split[8].

To train *wd1++* for evaluating on MATH500, we use dataset provided by (He et al., 2025), which is subsampled from OpenR1 dataset Face (2025). To evaluate on GSM8k, we leverage its train split to conduct *wd1++* training. Notably, we leverage a more effective system prompt and Math-Verify (Kydlíček) to parse the answers for full-parameter fine-tuning of *wd1*, *wd1++* and MDPO.

## B.2 Reward Function

To train *wd1* and reproduce *d1*, we use the reward function defined in (Zhao et al., 2025). For completion, we provide the details as following.

**GSM8K.** Following the Unsloth reward setup[9], we apply five addtive components: XML Structure Reward: +0.125 per correct tag; small penalties for extra content post-tags. Soft Format Reward: +0.5 for matching the pattern `<reasoning>...</reasoning><answer>...</answer>`. Strict Format Reward: +0.5 for exact formatting with correct line breaks. Integer Answer Reward: +0.5 if the answer is a valid integer. Correctness Reward: +2.0 if the answer matches ground truth.

**Countdown.** We include three cases: +1.0 if the expression reaches the target using the exact numbers. +0.1 if numbers are correct but target is missed. 0 otherwise.

**Sudoku.** The reward is the fraction of correctly filled empty cells, focusing on solving rather than copying.

**MATH500.** We include two additive subrewards. Format Reward is +1.00 for `<answer>` with `\boxed` inside; +0.75 for `<answer>` without `\boxed`; +0.50 for `\boxed` only. +0.25 for neither. Correctness Reward: +2.0 if the correct answer is in `\boxed{}`.

To train *wd1++*, we leverage Math-Verify (Kydlíček), constructing a simple verifier reward function to evalaute on GSM8K and MATH500.

## B.3 Sampling from Geometric Mixture

Although the sampling strategy eliminates the need to approximate the reference policy's likelihood, it incurs computational overhead, as generating a full completion requires multiple forward passes through the dLLM—compared to a single pass for likelihood estimation. An alternative is to sample from $\pi_{old}$ and shift the advantage to $\hat{A}_i = A^{\pi_{old}}(q, o_i) + \beta \log \hat{\pi_{ref}}/(\lambda + \beta)$, which reintroduces the need for reference policy likelihood approximation. However, policy ratio has been removed, and the reference model can be reused when conducting multiple gradient updates with the same batch of rollouts (off-policy). The increased computational burden is slight.

## B.4 Hyperparameters

We provide the hyperparameters of SFT in Table 5 and for *wd1* in Table 6.

|  | bacth_size | max_length | learning_rate | grad_accum_steps |
|---|---|---|---|---|
| **Value** | 1 | 4096 | 1e-5 | 4 |

Table 5: Hyperparameters of SFT in *d1* reproduction.

---

[5] `https://huggingface.co/datasets/openai/gsm8k`
[6] `https://huggingface.co/datasets/Jiayi-Pan/Countdown-Tasks-3to4`
[7] `https://github.com/Black-Phoenix/4x4-Sudoku-Dataset`
[8] `https://huggingface.co/datasets/ankner/math-500`
[9] `https://unsloth.ai/blog/r1-reasoning`

| Parameter | *wd1* | *d1* |
|---|---|---|
| **Model and Precision** | | |
| use_peft | true | true |
| torch_dtype | bfloat16 | bfloat16 |
| load_in_4bit | true | true |
| attn_implementation | flash_attention_2 | flash_attention_2 |
| lora_r | 128 | 128 |
| lora_alpha | 64 | 64 |
| lora_dropout | 0.05 | 0.05 |
| peft_task_type | CAUSAL_LM | CAUSAL_LM |
| **Training Configuration** | | |
| seed | 42 | 42 |
| bf16 | true | true |
| sync_ref_model | True | True |
| ref_model_sync_steps | 64 | 64 |
| adam_beta1 | 0.9 | 0.9 |
| adam_beta2 | 0.99 | 0.99 |
| weight_decay | 0.1 | 0.1 |
| $\psi$ (Equation (9)) | 1.0 | |
| max_grad_norm | 0.2 | 0.2 |
| warmup_ratio | 0.0001 | 0.0001 |
| learning_rate | 3e-6 | 3e-6 |
| lr_scheduler_type | constant_with_warmup | constant_with_warmup |
| **Batching and Evaluation** | | |
| per_device_train_batch_size | 6 | 6 |
| per_device_eval_batch_size | 1 | 1 |
| gradient_accumulation_steps | 2 | 2 |
| **RL** | | |
| num_generations | 6 | 6 |
| max_completion_length | 256 | 256 |
| max_prompt_length | 200 | 200 |
| block_length | 32 | 32 |
| diffusion_steps | 128 | 128 |
| generation_batch_size | 6 | 6 |
| remasking | low_confidence | low_confidence |
| random_masking | True | True |
| p_mask_prompt | 0.15 | 0.15 |
| beta | 0.00 | 0.04 |
| epsilon | – | 0.5 |
| num_iterations | 12 | 12 |

Table 6: Comparison of hyperparameters between *wd1* and *d1*.

### B.5 TRAINING COST ESTIMATION

For the runtime measurements reported in Table 2, we set $\mu = 8$ and train for a total of 6 global steps, corresponding to 48 gradient update steps. We use a batch size of 4 and the rest of the hyperparameters are the same as in Table 6. To estimate the number of function evaluations (NFEs) involved in computing likelihood approximations, we count only the forward passes, as the number of backward passes remains consistent across methods. The additional NFEs observed in the *d1* model arise from evaluating the likelihood under both the old and reference models, which are used for regularization. These extra evaluations are required only when new samples are drawn, as their outputs can be cached and reused across all gradient updates for $\mu$. We additionally report the number of floating-point operations (FLOPs) per global training step, measured using the Flops Profiler from Rasley et al. (2020).

### B.6 COMPUTING RESOURCES

For both *wd1* and *d1*, RL training is conducted on four NVIDIA A100 GPUs (80GB), and SFT is performed on four A6000 GPUs (48GB). For *wd1++* and MDPO, RL training is conducted on 8×A800 (80GB).

## C  ADDITIONAL EXPERIMENTS

We additionally report results for comparison to the results of the baseline *d1* reported in the paper (Zhao et al., 2025). As shown in Table 7, our method *wd1* evaluated and selected from less checkpoints, can outperform *d1* with a large margin in Sudoku and Countdown, achieving comparable performance in math problem-solving tasks.

### C.1  SUMMARY OF *wd1* RESULTS

Table 7: Test accuracy across different tasks. Our method demonstrates higher accuracy, especially significant in Sudoku and Countdown. The shaded area indicates where our method outperforms.

| Model | Sudoku | | Countdown | | GSM8K | | MATH500 | |
|---|---|---|---|---|---|---|---|---|
| | 256 | 512 | 256 | 512 | 256 | 512 | 256 | 512 |
| LLaDA-8B-Instruct | 6.7 | 5.5 | 19.5 | 16.0 | 76.7 | 78.2 | 32.4 | 36.2 |
| + diffu-GRPO *(reported)* | 12.9 | 11.0 | 31.3 | 37.1 | 79.8 | 81.9 | 37.2 | 39.2 |
| + diffu-GRPO *(reproduced)* | 16.1 | 11.7 | 27.0 | 34.0 | 80.7 | 79.1 | 34.4 | 39.0 |
| *d1 (reported)* | 16.7 | 9.5 | 32.0 | 42.2 | **81.1** | 82.1 | **38.6** | **40.2** |
| *d1 (reproduced)* | 17.6 | 16.2 | 25.8 | 35.2 | 78.2 | 82.0 | 34.4 | 38.0 |
| *wd1* | **76.4** | **62.8** | **51.2** | **46.1** | 80.8 | **82.3** | 34.4 | 39.0 |

Table 8: **Training cost.** The training steps to obtained the best post-trained model of three methods are 20, 150, and 7500. To compute the total rollouts, we need to compute the average rollouts in a single training step. Gradient steps per rollout batch represents the number of gradient descent conducted with a single batch of rollouts. In other words, 1 represents it is a pure on-policy RL training, and for any value $> 12$, off-policy RL is executed. Total Batch Size is computed by multiplying per-device batch size, gradient accumulation and the number of gpus. Therefore, the average number of rollouts used for single step gradient descent should be computed by total batch size divided by gradient steps per rollout batch.

| Hyperparameter | *wd1++* | MDPO | *d1* |
|---|---|---|---|
| Training step of the best checkpoint | 20 | 150 | 7500 |
| Training Steps per Rollout batch | 1 | 1 | 12 |
| Per-Device Batch Size | 4 | 1 | 6 |
| Gradient Accumulation | 2 | 16 | 2 |
| GPUs used for training | 8 | 8 | 4 |
| Total Batch Size | 64 | 128 | 48 |
| Avg. Rollouts per Step | 64 | 128 | 4 |
| Total Rollouts | 1280 | 19200 | 30000 |

We additionally provide reward dynamics in comparison to *wd1*-SFT in training. In Sudoku and Countdown, directly training with *wd1* without SFT shows significantly more efficient and stable learning process. In GSM8k and MATH500, the difference is negligible.

### C.2  ADDITIONAL ABLATION STUDY

We provide additional ablation study on the combined weight to confirm our analysis that the positive and negative samples terms in the loss function should be assigned equal proportion, due to the side case of a batch of all-negative generated responses (see the paragraph below Equation (9)). Assigning equal proportions to positive and negative weights is not arbitrary but rather the most robust design. This can be understood through two critical failure modes that arise from imbalanced proportions:

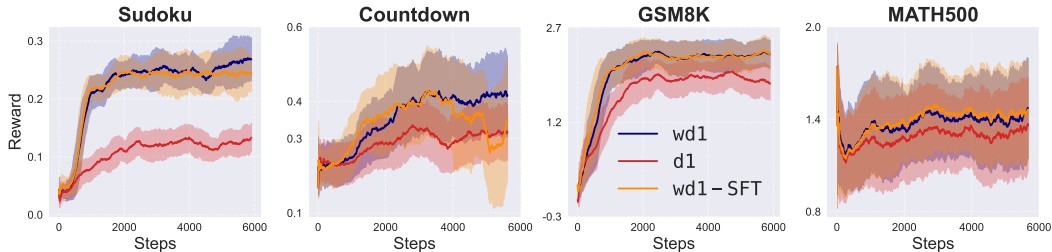

Figure 3: Reward Dynamics. *wd1* without SFT demonstrates better rewards in Sudoku and Countdown.

Table 9: Ablation on weight $\lambda$ to combine positive $w^+$ and negative weights $w^-$ in *wd1* on Sudoku. Specifically, the final weight assigned to log-likelihood is computed as $-\lambda w^+ + (1 - \lambda)w^-$.

| Combined Weight | Accuracy | Effective Tokens |
|---|---|---|
| 0.5 | 25.63% | 326.97 |
| 0.4 | 11.77% | 240.04 |
| 0.6 | 14.11% | 220.13 |

- When positive weight has larger proportion: In scenarios where all sampled completions have uniformly low rewards, a larger proportion of positive weights would paradoxically increase the log-likelihood of negative samples during wd1 optimization, which is clearly undesirable and contradicts the learning objective.

- When negative weight has larger proportion: Conversely, when all generated completions achieve uniformly high rewards, an insufficient proportion of positive weights would result in unlearning high-quality samples.

To empirically validate this analysis, we conducted experiments on the Sudoku dataset with varying mixing proportions. The results, presented in the table below, confirm our theoretical predictions.

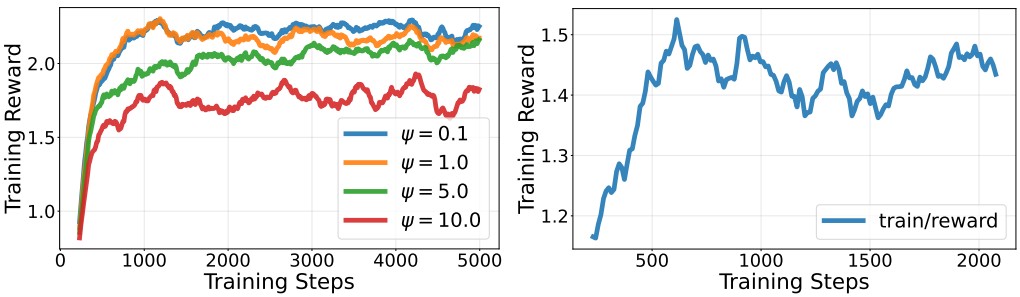

Figure 4: **Left:** Ablation study on $\psi$ in the weights (Equation (9)) on GSM8K. **Right:** *wd1* training on MATH with a random seed different from the seed used in our main experiments. The abrupt decrease of the rewards in the early training (see Figure 3 MATH500) disappears.

In all benchmark evaluations, we fix the hyperparameter $\psi = 1$, which controls the scale of the exponential weighting in *wd1*. To validate this choice, we provide an ablation study on the coefficient $\psi = 1$ in the exponential weight of *wd1* (Equation 9) below. Larger values $\psi$ leads to more extreme weight assigned to the samples. According to Figure 4, the training of applying different $\psi$ converges to similar rewards if $\psi$ is small. Overly large value (e.g. 10) can cause performance drop, implying that extreme weight assignment is detrimental.

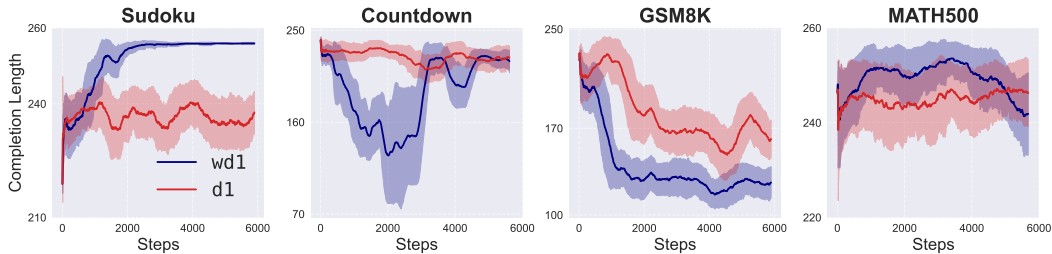

Figure 5: Completion lengths dynamics of *wd1* and *d1*. In math problem-solving tasks (GSM8K and MATH500), our method demonstrates smaller completion lengths and better token efficiency.

## C.3 CODING BENCHMARKS

We conduct 200 steps of *wd1* training on AceCode-87K (Zeng et al., 2025) following the implementation of Open-R1 (Hugging Face, 2025). Our method achieves consistent improvements over the base model.

Table 10: Comparative performance of *wd1* improvements compared to base model LLaDA-8B-Instruct. We present the results of *wd1* fine-tuned with AceCode dataset (Zeng et al., 2025).

| Task | Gen Length | Steps | Block Size | *wd1* | LLaDA |
|------|-----------|-------|-----------|-------|-------|
| HumanEval | 256 | 128 | 32 | **34.76** (+3.66) | 31.10 |
| HumanEval | 256 | 256 | 32 | **39.02** (+1.82) | 37.20 |
| HumanEval | 512 | 512 | 32 | **36.59** (+0.61) | 35.98 |
| MBPP | 128 | 128 | 32 | **39.2** (+2.40) | 36.8 |
| MBPP | 256 | 256 | 32 | **36.6** (+1.00) | 35.6 |
| MBPP | 512 | 512 | 32 | **36.8** (+0.40) | 36.4 |

## C.4 TRAINING DYNAMICS

Figure 3 presents the reward dynamics over gradient steps during training. *wd1* exhibits a notably faster reward increase compared to *d1*, highlighting its superior sample efficiency–effectively leveraging the reward signal to accelerate policy optimization. In addition, Figure 5 shows the average length of generated completions during training. On math reasoning benchmarks such as GSM8K and MATH500, *wd1* converges to shorter output sequences than *d1*, suggesting improved token efficiency while maintaining or improving performance.

## D LIMITATIONS

Similar to other RL-based approaches, *wd1* may lose effectiveness when all generations within a sampled group receive identical rewards. This situation can occur under several conditions—for example, when the training dataset is either too simple or too challenging for the base model. Nonetheless, such cases can be mitigated through careful reward design and the incorporation of curriculum learning strategies.

An additional limitation of this work is that the current *wd1* framework is restricted to text-based reasoning. Extending it to multimodal reasoning or unified diffusion-based models (e.g., (Yang et al., 2025)) represents a valuable direction for future research.

A final limitation concerns the likelihood approximation used in *wd1*. Our approach relies on the *d1*-based approximation, which is computationally efficient but introduces bias. Although some prior works employ ELBO-based estimators (e.g., DCE), they require additional computational overhead (Zhao et al., 2025) often exhibit high variance, as demonstrated in Figure 1. This trade-off highlights an important area for further exploration.

## D.1 ADDITIONAL ANALYSIS ON UNLEARNING

We provide extended demonstrations for Remark Remark 2, focusing specifically on the theoretical insights underlying the interpretation of the negative-sample reinforcement term in *wd1* as a form of data unlearning. Under the DCE likelihood approximation, the negative-sample reinforcement term in *wd1* becomes

$$\mathbb{E}_{o \sim p_0'(\cdot)}\left[w^-(q,o) \cdot \log \pi_\theta(o|q)\right] = \mathbb{E}_{o \sim p_0'(\cdot)}\left[\exp(-A(x_0)) \cdot \log \pi_\theta(o|q)\right] \tag{67}$$

$$=\mathbb{E}_{x_0 \sim p_0'(\cdot)}\left[\exp\big(-A(x_0)\big) \cdot \underbrace{\mathbb{E}_{t \sim [0,T], p_{t|0}'(x_t|x_0)}\Big[\sum_{x_t^i=[\mathtt{mask}]} -\frac{1}{t}\log p_\theta(x_0^i|x_t^{\mathrm{UM}})\Big]}_{\mathcal{L}_{\mathrm{DCE}}}\right] \tag{68}$$

$$=\mathbb{E}_{x_0^- \sim p_{\mathrm{data}}}\left[\underbrace{\mathbb{E}_{t \sim [0,T], p_{t|0}'(x_t|x_0^-)}\Big[\sum_{x_t^i=[\mathtt{mask}]} -\frac{1}{t}\log p_\theta(x_0^{-,i}|x_t^{\mathrm{UM}})\Big]}_{\mathcal{L}_{\mathrm{DCE}} \Leftrightarrow \mathrm{ELBO}}\right], \tag{69}$$

where $p_{\mathrm{data}}(x_0^-) = p_0'(x_0^-)\frac{\exp(-A(x_0^-))}{\sum_{x_0^-} p_0'(x_0^-)\exp(-A(x_0^-))}$. Equation (69) holds by simply applying importance sampling.

Since DCE is equivalent to the evidence lower bound (ELBO) of masked discrete diffusion models, we draw an analogy between the final objective in Equation (69) and data unlearning in diffusion models (Alberti et al., 2025). Equation (69) can be viewed as a direct masked discrete–diffusion extension of NegGrad (Golatkar et al., 2020), which aims to minimize the evidence lower bound of the log-likelihood on samples with lower advantage.

