# OpenReview forum: "wd1:  Weighted Policy Optimization for Reasoning in Diffusion Language Models"
_ICLR.cc/2026/Conference — ICLR 2026 Poster_

### Official Review · Reviewer_irEQ · 2025-10-26

**Soundness:** 3
**Presentation:** 3
**Contribution:** 3
**Rating:** 6
**Confidence:** 3

**Summary:**

This paper presents a weighted log-likelihood objective function for fine-tuning dLLMs via reinforcement learning. Compared to other RL-based approaches, the proposed method, *wd1*, improves the reasoning ability of dLLMs and also reduces the computational overhead due to the ratio-free policy optimization. In addition, the authors provide a theoretical insight to connect their method with energy-guided diffusion sampling, by viewing the energy function as the negative advantage. In experiments, the proposed method is evaluated on four reasoning benchmarks.

**Strengths:**

- The motivation and the main idea is clear, and the overall paper is well written and organized.
- The proposed weighted log-likelihood objective is interesting.
- In experiments, the proposed method outperforms other RL-based fine-tuning approaches.

**Weaknesses:**

1. The connection between the proposed method and the energy-guided diffusion model looks good but overstated. And the motivation is confusing: if fine-tuning the dLLM with wd1 is equivalent to training a energy-guided diffusion model, why don't you just fine-tuning dLLMs with the energy-based objective? The authors probably need to provide an ablation study or analysis on it.

2. In experiments, the proposed method is built on LLaDA and only evaluated on four benchmarks. But as a large language model, it should be also evaluated on more datasets such as those in the LLaDA paper (Table 2) to illustrate its scalability.

3. It would be better to give more discussion and related works for the “ratio-free” policy optimization. In Table 2, the difference of the training cost between two approaches is actually small. What's other advantages of using this “ratio-free” policy optimization?

4. This RL approach designs different reward functions for different tasks and datasets, which would be unpractical for a foundation LLM. And the proposed model is trained individually on each dataset, which might be unfair to LLaDA model because it is not specifically fine-tuned on these datasets.

**Questions:**

Please see "Weaknesses" and, what's the main limitation of this work? It would be better to provide a discussion on it.

---

> ### Author Response · Authors · 2025-11-20
> **Response to Reviewer irEQ**
>
> We thank the Reviewer for the constructive and positive feedback and appreciate the time and effort in reviewing our manuscript. We provide detailed responses to all questions as follows.
>
> &nbsp;
>
> > Q1: The connection between wd1 and the energy-guided diffusion model looks overstated. If wd1 is equivalent to training an energy-guided diffusion model, why not use energy-based objectives?
>
> **A1**: We agree that the statement of the equivalence is less rigorous, so we adjust it to ***“our proposed method can be interpreted as energy-guided discrete diffusion training combined with data unlearning”***. For better clarity, we have revised lines 313-323 in the manuscript for this question and provided detailed explanations as follows.
>
> As concluded in Remark 1, wd1 with only positive weights (i.e., WLL) is equivalent to the energy-based objective (AW-DCE, Eq. 14), when DCE (ELBO) is applied for likelihood approximation. We have included an ablation study in Table 4 by training only with WLL (i.e,. wd1-P). The results demonstrate that **training with positive weights alone leads to substantial performance degradation**, validating the necessity of our negative weighting mechanism. We have added the notation "(WLL)" after wd1-P in Table 4 in the new manuscript to make this connection explicit.
>
> Additionally, as stated in Remark 2, the negative weight term added to WLL to form wd1 can be viewed as performing **unlearning on low-advantage data** on top of energy-guided model training. As noted by reviewer *cbic*, this term effectively addresses a long-standing issue in weighted behavior-cloning methods (e.g., AWR) that samples are not fully utilized because the exponential weighting can become extreme. With the unlearning term, wd1 effectively leverages the low-advantage data to obtain a negative gradient, thus facilitating policy optimization.
>
> &nbsp;
>
> > Q2: More datasets should be included.
>
> **A2**: **We conduct new experiments on coding benchmarks.** The wd1 training is conducted on AceCode-87K. Our method achieves consistent improvements over the base model.
>
> |Task    | Gen Length | Steps | Diffu Block | $\quad$ **wd1** $\quad$ | LLaDA-8B-Instruct |
> |-----------|:------------:|:------:| :------------:|:-------------:|:-------------:|
> | HumanEval | 256 | 128 | 32 | **34.76** | 31.10 |
> | HumanEval  | 256 | 256 | 32 | **39.02** | 37.20 |
> | MBPP       | 128 | 128 | 32 | **39.2**  | 36.8 |
> | MBPP        | 256| 256| 32 | **36.6**  | 35.6 |
>
> &nbsp;
>
> > Q3: The difference in the training cost between d1 and wd1 is actually small. What's other advantages of using this “ratio-free” policy optimization?
>
> **A3**:
>
> **1. Training cost**
>
> We respectfully disagree with the reviewer's assessment that the training cost difference between d1 and wd1 is small. We wish to clarify several critical points of wd1’s advantages:
>
> **Substantial Cost Savings at Scale:** The RL training costs reported in Table 2 represent **per-step costs**. When scaled to realistic training regimes of thousands or tens of thousands of optimization steps—standard practice in RL fine-tuning—the cumulative cost savings become **substantial**.
>
> **Further Reduction in On-Policy Settings:** The reported costs assume a fixed configuration of 8 inner gradient steps per training iteration. However, for purely on-policy training (i.e., \\mu=1), wd1 achieves an even more dramatic reduction, requiring only **⅓ of the NFEs compared to d1**. This demonstrates that the efficiency gains are not merely incremental but constitute a fundamental algorithmic advantage.
>
>
> **2. Other Advantage**
>
> **Superior Performance.** We would like to emphasize that the very first advantage is superior performance, especially on planning tasks, countdown, and sudoku, as presented in Table 1\.
>
> **Addressing the Ratio Approximation Challenge:** Beyond computational efficiency, the ratio-free formulation of wd1 addresses a fundamental methodological limitation elaborated in Figure 1 and lines 131-146. Specifically, the likelihood ratios computed via ELBO-based approximation suffer from two critical issues: (1) **systematic bias** when using small sample sizes of timesteps t, and (2) **high variance** in the estimates. The ratio-free formulation of wd1 **completely circumvents these challenges**, eliminating both the bias and variance issues inherent to likelihood-based ratio estimation.
>
> In summary, the advantages of wd1 over d1 encompass addressing fundamental challenges of ratio estimation, as well as reducing substantial computational cost at scale in practice.

---

> ### Author Response · Authors · 2025-11-20
> **Response to Reviewer irEQ**
>
> > Q4: It would be better to give more discussion and related works for the “ratio-free” policy optimization.
>
> **A4**: Related work on ratio-free policy optimization has been covered in Section 6 under "RL via Weighted Regression," discussing methods that use weighted log-likelihood objectives, thus being ratio-free. We further expand the discussion on ratio-free policy optimization as follows, and add it to Appendix E.
>
> If a policy optimization objective requires neither importance sampling nor regularization with respect to a reference model, then the objective is ratio-free. Consequently, on-policy algorithms such as vanilla policy gradient methods (e.g., REINFORCE) and their variants (e.g., RLOO) are inherently ratio-free. This property is particularly valuable for dLLMs, where errors in log-likelihood approximation can accumulate and propagate through ratio-based computations. Concurrent work, such as SPG \[5\], adopts a policy-gradient formulation and develops an objective tailored specifically for diffusion language models. Another on-policy optimization approach, d2 \[6\], removes both the marginal likelihood ratios and the likelihood terms from the RL objective for dLLMs, offering a more fundamental solution.
>
> &nbsp;
>
> > Q5: This RL approach designs different reward functions for different tasks and datasets, which would be impractical for a foundation LLM. The proposed model is trained individually on each dataset, which might be unfair to the LLaDA model because it is not specifically fine-tuned on these datasets.*
>
> **A5**: We respectfully disagree with the claim that task-specific reward functions make our approach impractical for foundation LLMs.
>
> The reward functions used in this paper—**accuracy reward and format reward**—represent **standard practice** in RL for reasoning tasks (see DeepSeek R1 [4]). Accuracy reward assigns a positive value for correct outputs or indicates the level of correctness. Format reward serves only to facilitate answer parsing. In any task, the completions should be evaluated from these two critical aspects. Thus, it’s, on the contrary, practical for any reasoning tasks for a foundation LLM. Besides, as demonstrated in line 1161, we leverage a simple task-agnostic verifier as the reward function for wd1++ training and achieve superior performance, further indicating the effectiveness of our approach.
>
> **LLaDA as Baseline:** wd1, as an RL post-training algorithm, is **not intended to compete with LLaDA**. Rather, LLaDA is included in Tables 1 and 3 to establish a clear baseline and demonstrate the **net contribution** that wd1 provides beyond the pretrained checkpoint. Since wd1 training begins from the LLaDA checkpoint, the appropriate comparisons are with **other RL methods (such as d1)** that similarly start from LLaDA and utilize the same training data. This experimental design allows for controlled evaluation of the RL algorithm for dLLMs itself, isolating its contribution from differences in base model quality or training data.
>
> **RL-based Post-Training:** The per-dataset fine-tuning experiments are designed to clearly demonstrate that wd1's performance gains are **robust and consistent across diverse task distributions**. Importantly, these experiments should not be interpreted as suggesting that RL methods must be applied to single datasets—this is merely an evaluation methodology to assess robustness across different data characteristics. In addition, this evaluation protocol has been widely adopted in RL for dLLM literature [1-3]. The algorithm is fully compatible with multi-task foundational training, and it is a valuable direction for **future work**.
>
> &nbsp;
>
> > Q6: Limitations
>
> We added a limitation section in Appendix D in the manuscript.
>
> ---
>
> &nbsp;
>
> We have carefully addressed each concern raised and are happy to provide any additional clarifications needed. We hope our comprehensive responses have resolved the reviewer's questions and demonstrated the value of our contributions. If so, we would be grateful if the reviewer would consider increasing the rating.
>
> &nbsp;
>
> ---
>
> [1] Zhao, Siyan, et al. "d1: Scaling reasoning in diffusion large language models via reinforcement learning." arXiv preprint arXiv:2504.12216 (2025).
>
> [2] Wang, Wen, et al. "Time is a feature: Exploiting temporal dynamics in diffusion language models." arXiv preprint arXiv:2508.09138 (2025).
>
> [3] He, Haoyu, et al. "Mdpo: Overcoming the training-inference divide of masked diffusion language models." arXiv preprint arXiv:2508.13148 (2025).
>
> [4] Guo, Daya, et al. "Deepseek-r1: Incentivizing reasoning capability in llms via reinforcement learning." *arXiv preprint arXiv:2501.12948* (2025).
>
> [5] Wang, Chengyu, et al. "Spg: Sandwiched policy gradient for masked diffusion language models." *arXiv preprint arXiv:2510.09541* (2025).
>
> [6] Wang, Guanghan, et al. "d2: Improved Techniques for Training Reasoning Diffusion Language Models." *arXiv preprint arXiv:2509.21474* (2025).

---

> ### Comment · Reviewer_irEQ · 2025-11-24
>
> I appreciate the authors' efforts in the rebuttal. However, I'm still confused by the statement of energy-guided diffusion. In the experiment (Table 4), *wd1* with only positive weights performs much worse than the standard *wd1* and even the original LLaDA-8B-Instruct. Does it mean the energy-guided diffusion objective is useless in LLaDA post-training? Then why the authors introduce it using a separate section and list it as an important contribution? And what if training wd1 with only negative sample reinforcement ($w^-$)?
>
> In addition, I still believe the training cost saving is incremental (and I know it's per-step costs). As a post-training algorithm, ~20% training cost saving is good but definitely not fundamental.
>
> In the rebuttal, the authors further claim that "the very first advantage is superior performance". But as far as I see in the revised paper, the improvements are obvious only on the *Sudoku* and *Countdown* benchmarks, both are actually unstable for large model evaluation. So the authors should at least also report the comparison results on the two tasks in Table 3. And it would be great to add an extensive evaluation across more tasks with the same comparison methods.
>
> Moreover, the new experiments on HumanEval and MBPP are not convincing since LLaDA is not trained/finetuned on the same dataset, and the official results of LLaDA-8B-Instruct on the two tasks are **49.4** and **41.0**, both are higher than the proposed wd1.
>
> The authors also mentioned DeepSeek R1 and its reward functions in A5. But DeepSeek R1 uses extensive human preference rewards in the final alignment phase and is evaluated on more than 20 tasks. For the proposed wd1, it remains questionable that if wd1 can be scaled with the same human preference rewards and can consistently outperform other methods across diverse tasks. Moreover, there is no discussion or experiment that can prove "The algorithm is fully compatible with multi-task foundational training".
>
> Overall, I believe this is a borderline paper to ICLR, and I look forward to the new response and further discussion with the authors.

---

> > ### Author Response · Authors · 2025-12-01
> > **Response to Reviewer irEQ**
> >
> > We thank the reviewer for the thorough and insightful feedback, as well as the willingness for further discussion\! We provide detailed explanations to further address confusion and misunderstanding.
> >
> > &nbsp;
> >
> > > Q7: Does it mean the energy-guided diffusion objective is useless in LLaDA post-training? Then why do the authors introduce it using a separate section and list it as an important contribution? What if training wd1 with only negative sample reinforcement?
> >
> > **A7:**
> >
> > **Energy-guidance objective is necessary because the objective without energy guidance, i.e. purely training wd1 with only negative samples can dramatically fail.** We have rerun experiments to evaluate objectives that only include negative samples, e.g., on the Sudoku benchmark. There is a significant performance drop, and the training becomes completely ineffective in the later stage. Please refer to Figure 5 in Appendix F in our revised manuscript. **This result clearly shows the necessity of adding energy-guidance (positive weight) in wd1, further confirming the effectiveness of our design choice—combining positive and negative weights.**
> >
> > Additionally, in the separate section 4, we not only aim to introduce energy guidance to interpret wd1 with positive weights (Remark 1), **but also further introduce direct interpretation of wd1 (positive \+ negative weight) via energy guidance \+ negative sample unlearning (Remark 2).** To help better understand the connection to unlearning, we provide an extended demonstration in Appendix F.1.
> >
> > We would like to finally clarify that the section introducing the energy-guidance objective and unlearning is our **theoretical contribution**, used to interpret the wd1 from a principled diffusion sampling perspective. In this section, we did not aim to propose any new objective but to set up theoretical connections to support our objective with grounded theory. AW-DCE introduced in this section is derived from guided sampling, a principled method in diffusion models (e.g. \[1,7\]). Thus, having this connection between RL and guided sampling for dLLMs helps confirm the theoretical soundness of wd1 and even RL (reverse-KL regularized policy optimization) in general.
> >
> > &nbsp;
> >
> > > Q8: As a post-training algorithm, \~20% training cost saving is good but definitely not fundamental.
> >
> > **A8:** We respectfully note that the assessment of what constitutes a "significant" improvement is inherently subjective. From our perspective, a **20% performance gain** achieved solely through a change in optimization objectives without additional engineering modifications or extensive hyperparameter tuning represents a **substantial contribution**. This demonstrates that principled algorithmic choices can yield meaningful improvements at minimal implementation cost.
> >
> > &nbsp;
> >
> > > Q9: The improvements are obvious only on the *Sudoku* and *Countdown* benchmarks, both are actually unstable for large model evaluation. And it would be great to add an extensive evaluation across more tasks with the same comparison methods.
> >
> > **A9:**
> >
> > ***Sudoku*** **and *Countdown* have been used in RL for dLLMs works that have already been published on NeurIPS 2025, including  LLaDOU \[2\] and d1 \[3\]**, as well as various follow-up works \[4,5,6\]. Therefore, we believe these two benchmarks are reasonable choices, and more importantly, we aim to compare with the baseline methods fairly. Thus, our evaluation setup **closely matches** the standard of previous published works of RL for dLLMs.
> >
> > &nbsp;
> >
> > > Q10-1: Moreover, the new experiments on HumanEval and MBPP are not convincing since LLaDA is not trained/finetuned on the same dataset.
> >
> > **A10-1:** We believe there may be a misunderstanding regarding the evaluation setup. The purpose of post-training is precisely to demonstrate improvement over the base model (in our case, LLaDA) using data that was not seen during pre-training. It is standard practice and indeed the intended experimental design that the base model has not been trained on the post-training dataset. Our results on HumanEval and MBPP thus follow the conventional evaluation protocol for assessing post-training methods.
> >
> > &nbsp;
> >
> > > Q10-2: And the official results of LLaDA-8B-Instruct on the two tasks are **49.4** and **41.0**, both are higher than the proposed wd1.
> >
> > **A10-2:** To replicate LLaDA-8B-Instruct’s results, we use their official codebase which contains the evaluation scripts, and their published checkpoints on huggingface. The default evaluation script in LLaDA used 1024 completion length and diffusion steps. However, due to limited time and compute, we have to only set 128/256 completion length and diffusion steps and instead got 37.20 and 36.8. Based on these LLaDA results, **wd1 evaluated with the same completion length and diffusion steps has demonstrated clear improvement.**

---

> > > ### Author Response · Authors · 2025-12-01
> > > **Response to Reviewer irEQ**
> > >
> > > > Q11: The authors also mentioned DeepSeek R1 and its reward functions in A5. But DeepSeek R1 uses extensive human preference rewards in the final alignment phase and is evaluated on more than 20 tasks. For the proposed wd1, it remains questionable if wd1 can be scaled with the same human preference rewards and can consistently outperform other methods across diverse tasks. Moreover, there is no discussion or experiment that can prove "The algorithm is fully compatible with multi-task foundational training".
> > >
> > > **A11:**
> > >
> > > Thanks for pointing this out. In light of limited computing resources, evaluating more than 20 tasks like DeepSeek R1 is impractical for us, especially since it is hard to conduct careful ablation studies (e.g., varying completion length, positive weights, SFT, etc.) in our paper. **However, as explained in A9, our evaluation setup matches the standard as in previous published works \[2-3\].** We also agree that multi-task foundational training would be an interesting extension in future work to apply our novel reweighting RL paradigm.
> > >
> > > ---
> > >
> > > We appreciate the reviewer's careful and constructive engagement with our work, which contributes to the inspiring discussion!
> > >
> > > &nbsp;
> > >
> > > [1] Ho, Jonathan, and Tim Salimans. "Classifier-free diffusion guidance." arXiv preprint arXiv:2207.12598 (2022).
> > >
> > > [2] Huang, Zemin, et al. "Reinforcing the diffusion chain of lateral thought with diffusion language models." arXiv preprint arXiv:2505.10446 (2025)
> > >
> > > [3] Zhao, Siyan, et al. "d1: Scaling reasoning in diffusion large language models via reinforcement learning." arXiv preprint arXiv:2504.12216 (2025)
> > >
> > > [4] Wang, Guanghan, et al. "d2: Improved Techniques for Training Reasoning Diffusion Language Models." arXiv preprint arXiv:2509.21474 (2025).
> > >
> > > [5] Wang, Chengyu, et al. "Spg: Sandwiched policy gradient for masked diffusion language models." arXiv preprint arXiv:2510.09541 (2025).
> > >
> > > [6] He, Haoyu, et al. "Mdpo: Overcoming the training-inference divide of masked diffusion language models." arXiv preprint arXiv:2508.13148 (2025).
> > >
> > > [7] Schiff, Yair, et al. "Simple guidance mechanisms for discrete diffusion models." arXiv preprint arXiv:2412.10193 (2024).

---

### Official Review · Reviewer_cbic · 2025-11-01

**Soundness:** 3
**Presentation:** 3
**Contribution:** 3
**Rating:** 6
**Confidence:** 3

**Summary:**

This paper introduces wd1, a novel reinforcement learning algorithm tailored for diffusion Large Language Models. Unlike traditional on-policy RL, wd1 leverages the closed-form optimal solution of the KL-regularized RL problem. It projects the current model toward this optimum using weighted regression on samples generated by behavior policies.

The authors also identify a significant issue regarding the under-penalization of negative samples and propose a complementary penalty term to address it. Experiments on popular math and reasoning benchmarks demonstrate that wd1 achieves significant improvements over the d1 baseline. Furthermore, ablation studies successfully validate the effectiveness of the algorithm's key design choices.

**Strengths:**

Wd1 employs a clever approach to bypass the computation of the log-probability ratio, which is often intractable or computationally expensive for diffusion LLMs.

The introduced penalty term for negative samples is a notable contribution. It appears to effectively address a long-standing issue in weighted behavior-cloning methods—namely, the insufficient penalization of negative samples. The proposed solution is both simple and effective.

**Weaknesses:**

wd1 appears highly sensitive to the coefficient that mixes positive weights w+ and negative weights w-. In Table 9, varying this coefficient lambda by 0.1 results in an accuracy drop of over 10%, which suggests the model may require careful tuning.

**Questions:**

In Figure 2 (MATH500), all tested RL algorithms seem to cause a significant drop in reward compared to the baseline. Could the authors elaborate on why this might be the case?

Some copy-edits: Lemma 1, A_t(x_0) should be A_t(x_t).

---

> ### Author Response · Authors · 2025-11-20
> **Response to Reviewer cbic**
>
> We thank the reviewer for the positive feedback and detailed review. We have corrected the typo $A\_t(x\_t)$ in the revised version. Below, we address the questions and concerns in detail.
>
> &nbsp;
>
> > Q1: wd1 appears sensitive to the coefficient mixing positive and negative weights, which implies the method requires careful tuning.
>
> **A1**: We would like to clarify that **the mixing proportion between positive and negative weights is not a hyperparameter.** Instead, we **enforce equal proportions** between positive and negative weights. Table 9 is included specifically to **empirically validate this theoretical analysis**, demonstrating that equal proportions emerge as the optimal configuration. We have discussed this in Section 3.2 (lines 223-227) in the original manuscript. For better clarity, we provide further detailed analysis below, and add it to Appendix C.2 in the revised manuscript.
>
> **Theoretical Justification for Equal Proportions:** Assigning equal proportions to positive and negative weights is not arbitrary but rather the **most robust design**. This can be understood through two critical failure modes that arise from imbalanced proportions:
>
> 1. **When positive weight has a larger proportion:** In scenarios where all sampled completions have uniformly low rewards, a larger proportion of positive weights would paradoxically **increase the log-likelihood of negative samples** during wd1 optimization, which is clearly undesirable and contradicts the learning objective.
> 2. **When negative weight has a larger proportion:** Conversely, when all generated completions achieve uniformly high rewards, an insufficient proportion of positive weights would result in **unlearning high-quality samples**.
>
> The equal proportion scheme elegantly avoids both pathological cases, ensuring robust learning across diverse reward distributions.
>
> &nbsp;
>
> > Q2: In MATH500, all tested RL algorithms dropped, why is that?
>
> **A2**: When we switch to a different random seed—used for shuffling the training data—the abrupt drop in the early training reward no longer appears (please refer to the right subfigure of Figure 3). This behavior may stem from the shuffled data presenting an unusually easy set of math problems in the early batches. Nevertheless, the test accuracy improves regardless of this issue, demonstrating that the RL procedure still effectively improves the model’s reasoning ability.
>
> &nbsp;
>
> ---
>
> We hope we have addressed all the reviewers' questions, particularly clarifying the misunderstanding regarding Question 1, which has helped us identify important improvements for the paper. In light of these clarifications, we would be grateful if the reviewer would consider increasing their score.

---

> > ### Comment · Reviewer_cbic · 2025-11-23
> >
> > I appreciate the response and the further clarification. All my concerns and questions have been resolved, and I will increase my evaluation.

---

> ### Author Response · Authors · 2025-11-24
> **Response to Reviewer cbic**
>
> Thank you so much for taking the time to reconsider and increase the score — We genuinely appreciate your thoughtful review and support.

---

### Official Review · Reviewer_kDWu · 2025-11-07

**Soundness:** 3
**Presentation:** 4
**Contribution:** 3
**Rating:** 8
**Confidence:** 3

**Summary:**

This paper introduces wd1 and wd1++, a novel RL algorithm for dLLMs reasoning finetuning. Instead of requiring likelihood approximation in previous work, wd1 is ratio-free and from weighted log-likelihood maximization. Such a formulation reduces the estimation variance and the compute cost. The paper provides a theoretical foundation and demonstrates the connection with energy-based model training. Empirically, wd1 and wd1++ demonstrates superior performance on four benchmarks, while requiring fewer rollouts and no SFT.

**Strengths:**

- The idea is well-motivated and paper is well-organized and easy to follow.,
- The paper has strong theoretical foundation and analysis.,
- The experiments are thorough and verify the effectiveness of the methods and design choices.,

**Weaknesses:**

- From my understanding, one key design choice is to choose reverse KL instead of forward KL, which makes the optimization analytic and ratio-free. However, the paper lacks the discussion on this motivation and its impact, e.g., how it will affect the constraint.,
- The limitations are not discussed in detail. I didn’t find any place to explicitly discuss the limitations of the method.,

**Questions:**

- Given the wd1 objective, is it easy to collapse to some certain solutions? how diversity the finetuned output is?,
- Regarding the exponential weights, how easy to tune the weight parameters? is the performance sensitive to the scale?

---

> ### Author Response · Authors · 2025-11-20
> **Response to Reviewer kDWu**
>
> We sincerely thank the reviewer for the thoughtful and encouraging evaluation. We are grateful for the positive assessment of our work's strengths and provide detailed responses to the raised questions below.
>
> &nbsp;
>
> > Q1: Reverse KL is the key, but the paper lacks the motivation and impact (e.g. on the constraint).
>
> **A1**: The impact (e.g. on policy update constraint) has been briefly demonstrated in Section 3.1.
>
> **Impact on constraint.** As discussed in lines 178-180, **policy optimization with reverse-KL penalty satisfies the monotonic improvement property established in TRPO.** This follows from the fact that the original policy update is constrained by total variation (TV) divergence (see [1] Theorem 1). While TRPO employs forward-KL as an upper bound of TV divergence, the symmetry of TV divergence ensures that reverse-KL also serves as a valid upper bound and can also be used for policy update constraints to ensure monotonic improvement.
>
> **Motivation.** Reverse-KL-constrained policy optimization (also known as policy mirror descent) is a well-established framework with demonstrated effectiveness across diverse settings. Notable applications include advantage-weighted regression (AWR) [1], diffusion-based RL [5], self-play alignment [6], and recent large-scale LLM post-training deployments such as Kimi K1.5 and K2 [2-3]. This extensive body of work provides strong empirical validation.
>
> Beyond this empirical support, our choice of reverse-KL is fundamentally motivated by its enabling of ratio-free policy optimization, as rigorously derived in Section 3.1. This is not merely a computational convenience but rather a **principled solution to a fundamental limitation** of applying standard RL methods to diffusion language models, where likelihood evaluation is intractable and approximations (e.g., via ELBO) introduce both bias and variance.
>
> &nbsp;
>
> > Q2: Limitations of wd1
>
> **A2**: We provide the limitation as follows and have included this limitation section in the new manuscript in Appendix D.
>
> Similar to other RL-based approaches, wd1 may lose effectiveness when all generations within a sampled group receive identical rewards. This situation can occur under several conditions—for example, when the training dataset is either too simple or too challenging for the base model. Nonetheless, such cases can be mitigated through careful reward design and the incorporation of curriculum learning strategies.
>
> An additional limitation of this work is that the current wd1 framework is restricted to text-based reasoning. Extending it to multimodal reasoning or unified diffusion-based models (e.g., \[7\]) represents a valuable direction for future research.
>
> A final limitation concerns the likelihood approximation used in wd1. Our approach relies on the d1-based approximation, which is computationally efficient but introduces bias. Although some prior works employ ELBO-based estimators (e.g., DCE), they often exhibit high variance as demonstrated in Figure 1\. This trade-off highlights an important area for further exploration.
>
> &nbsp;
>
> > Q3: Is wd1 easy to collapse and reduce diversity?
>
> **A3**: While RL algorithms as reward maximization frameworks generally reduce output diversity—particularly those employing reverse-KL regularization, which are inherently mode-seeking—**wd1’s weighted log-likelihood objective actively mitigates this tendency**. Specifically, because wd1 employs softmax-based weighting in its loss function, the log-likelihoods of **all positive samples are maximized simultaneously**, rather than converging to a single mode. This creates a **multi-modal distribution** over the high-reward region, thereby preserving diversity among high-quality outputs.

---

> > ### Author Response · Authors · 2025-11-20
> > **Response to Reviewer kDWu**
> >
> > > Q4: Is it easy to tune the hyperparameter in the exponential weight?
> >
> > **A4**: We additionally include an ablation study on the coefficient psi in the exponential weighting scheme of wd1 (please refer to Figure 3 in Appendix C.2 in the new manuscript). As shown in the figure, training with different psi values converges to similar reward levels when psi is small. However, excessively large values (e.g., 10\) lead to noticeable performance degradation, indicating that overly extreme weighting is detrimental.
> >
> > ---
> >
> > \[1\] Peng, Xue Bin, et al. "Advantage-weighted regression: Simple and scalable off-policy reinforcement learning." *arXiv preprint arXiv:1910.00177* (2019).
> >
> > \[2\] Team, Kimi, et al. "Kimi k1.5: Scaling reinforcement learning with LLMs." *arXiv preprint arXiv:2501.12599* (2025).
> >
> > \[3\] Team, Kimi, et al. "Kimi k2: Open agentic intelligence." *arXiv preprint arXiv:2507.20534* (2025).
> >
> > \[4\] Cui, Ganqu, et al. "The entropy mechanism of reinforcement learning for reasoning language models." *arXiv preprint arXiv:2505.22617* (2025).
> >
> > \[5\] Ma, Haitong, et al. "Efficient Online Reinforcement Learning for Diffusion Policy." *arXiv preprint arXiv:2502.00361* (2025).
> >
> > \[6\] Wu, Yue, et al. "Self-play preference optimization for language model alignment." *arXiv preprint arXiv:2405.00675* (2024).
> >
> > \[7\] Yang, Ling, et al. "Mmada: Multimodal large diffusion language models." *arXiv preprint arXiv:2505.15809* (2025).

---

### Author Response · Authors · 2025-12-03
**General Response to AC**

Dear AC,

We sincerely appreciate the time and effort the AC has devoted to evaluating our manuscript. We obtained **initial scores 8, 6, and 6**. We have made significant efforts to address all questions and concerns raised by the reviewer, including providing further explanations, analyses, and supplementary experiments. In the following, we would like to summarize our discussions with reviewers.

&nbsp;

**Reviewer-kDWu**

The reviewer **rated our manuscript an 8**\. We provided careful explanations in response to their comments and were awaiting further feedback. The main concerns are as follows:

1. The reviewer found it unclear about the motivation and impact of applying reverse KL as a constraint for the policy update.
   - We clarified that these points are discussed in Section 3.1: reverse KL enables an effective simplification of the objective (motivation), while the monotonic improvement guarantee established in TRPO still holds (impact).
2. The reviewer also raised questions regarding the hyperparameter $\\psi$ in the exponential weighting scheme and the resulting response diversity.
   - We provided additional experiments and explanations demonstrating the robustness of the method to the choice of $\\psi$, as well as the diversity introduced by the softmax weighting.

&nbsp;

**Reviewer-cbic**

The reviewer had two concerns, which were quickly and successfully resolved by our responses. As a result, **the reviewer raised their evaluation from 6 to 8, making the score 8, 8, 6\.**

- The first concern was wd1’s sensitivity to the mixing weight of the negative samples. We clarified that the weight of the negative samples **is not a hyperparameter** in wd1. Equal weighting on positive and negative samples **is theoretically enforced** and also yields the best performance.
- The second concern was about the reward drop at the beginning of the training reward curve of MATH500. We clarified that this was due to data shuffling, and the decreasing trend disappears when we change to another random seed.

&nbsp;

**Reviewer-irEQ**

The reviewer initially voted for **6** and raised three major concerns, which were addressed thoroughly in our response. We are confident that, had the discussion continued, we would have fully resolved all the issues, likely resulting in an improved score.

The concerns and our responses can be summarized as follows:

1. Misunderstanding on our interpretation of wd1 with energy-guided diffusion, and the goal of section 4.
   - We further polished section 4, and clarified that this section includes the theoretical interpretation of our approach. Our manuscript also contains an ablation study showing that energy guidance is crucial in wd1.
2. Need for experiments with more datasets.
   - We provided new results on coding benchmarks, HumanEval, and MBPP, showing that wd1 consistently improves performance.
3. Need for additional discussion on ratio-free policy optimization.
   - We expanded related work in Appendix E, providing more works without importance sampling are on-policy algorithms. However, our method supports off-policy training.

&nbsp;


Once again, we extend our sincere gratitude for the time, effort, and careful attention you have devoted to reviewing our work.

Best regards,

The authors

---

### Meta-Review · Area_Chair_KbsH · 2026-01-05

**Summary:**

This paper introduces wd1, a ratio-free policy optimization framework tailored for diffusion LLMs (dLLMs) that circumvents the intractable likelihood ratio problem found in prior work. The submission received positive initial support (Scores: 8, 6, 6). Reviewers consistently recognized the novelty of the ratio-free policy optimization framework. Key discussion points included:

(1) Theoretical Grounding: The justification for Reverse-KL constraints and the connection to energy-guided diffusion.

(2) Hyperparameter Sensitivity: Concerns about the stability of mixing weights and temperature parameters.

(3) Empirical Rigor: Questions regarding efficiency gains, baseline fairness (vs. official LLaDA results), and generalization to non-math domains.

I recommend acceptance. The authors provided a solid rebuttal that resolved the primary technical concerns regarding stability.

**Reviewer Concerns:**

Addressed Concerns:

- Sensitivity of Mixing Weights (Reviewer cbic): Concerns regarding the stability of mixing weights (Table 9) were resolved. The authors clarified that equal weighting is not merely a heuristic but is theoretically enforced and empirically optimal.
- Reverse-KL Motivation (Reviewer kDWu): The authors substantiated the choice of Reverse-KL with appropriate theoretical references.
- Generalization (Reviewer irEQ): The authors added experiments on coding benchmarks to demonstrate applicability beyond math tasks.

Outstanding Concerns:

- Baseline Fairness & Cost (Reviewer irEQ): Reviewer irEQ remains skeptical about the significance of the ~20% cost reduction, viewing it as incremental. Additionally, they noted that the reproduced LLaDA baselines trail official reports (attributed to compute constraints). While the relative improvement is acknowledged, the absolute gap compared to SOTA remains an issue.
- Theoretical Interpretation (Reviewer irEQ): Despite the rebuttal, Reviewer irEQ remained unconvinced by the "energy-guided diffusion" connection and questioned the utility of the energy-based objective in the post-training context.

**Reviewer Scores:**

- Reviewer kDWu (Initial: 8): 8 (Maintain).
- Reviewer cbic (Initial: 6): 8 (Increase).
- Reviewer irEQ (Initial: 6): 6 (Maintain). Although the reviewer appreciated the effort, they explicitly stated the paper is "borderline" due to unconvincing experiments and the perceived incremental nature of the efficiency gains.

---

### Decision · Program_Chairs · 2026-01-26

Accept (Poster)